# Adaptive Symmetrization of the KL Divergence

## Abstract

Many tasks in machine learning can be described as or reduced to learning a probability distribution given a finite set of samples. A common approach is to minimize a statistical divergence between the (empirical) data distribution and a parameterized distribution, e.g., a normalizing flow (NF) or an energy-based model (EBM). In this context, the forward KL divergence is a ubiquitous due to its tractability, though its asymmetry may prevent capturing some properties of the target distribution. Symmetric alternatives involve brittle min-max formulations and adversarial training (e.g., generative adversarial networks) or evaluating the reverse KL divergence, as is the case for the symmetric Jeffreys divergence, which is challenging to compute from samples. This work sets out to develop a new approach to minimize the Jeffreys divergence. To do so, it uses a proxy model whose goal is not only to fit the data, but also to assist in optimizing the Jeffreys divergence of the main model. This joint training task is formulated as a constrained optimization problem to obtain a practical algorithm that adapts the models priorities throughout training. We illustrate how this framework can be used to combine the advantages of NFs and EBMs in tasks such as density estimation, image generation, and simulation-based inference.

## 1 Introduction

An established approach to learn from observations is to assume they are drawn from a probability distribution (Murphy, 2012) and find the best distribution that explains the data. By learning the underlying distribution we can solve tasks such as classification, regression, and sampling (Kingma & Welling, 2022; LeCun et al., 2015; Grathwohl et al., 2020). Parameterized functions in the form of neural networks are a pivotal instrument to bring this theory into practice, as they can model complex distributions (Hornik, 1991). Training a network to capture a distribution typically involves minimizing the difference between the ground truth distribution, accessible only through a finite number of samples, to the distribution represented by the network. For instance, parameterized distributions such as normalizing flows (NFs) (Tabak & Vanden-Eijnden, 2010; Tabak & Turner, 2013; Papamakarios et al., 2021) or energy-based models (EBMs) (Teh et al., 2003; Du & Mordatch, 2019) are generally trained minimizing the *forward* Kullback-Leibler (KL) divergence $D_{\mathrm{KL}}(\pi \parallel p)$ between the true data distribution $\pi$ and the model distribution $p$. However, because of the asymmetry of the KL divergence, i.e., $D_{\mathrm{KL}}(\pi \parallel p) \neq D_{\mathrm{KL}}(p \parallel \pi)$, training on discrete samples may lead to a mismatch between the modeled distribution and the data distribution (illustrated in Figure 1a). Ordinarily, minimizing a symmetric divergence would alleviate this issue, but it demands knowing the data distribution directly. Generative adversarial networks (GANs) (Goodfellow et al., 2014; Arjovsky et al., 2017) overcome this obstacle by representing a symmetric divergence as a variational problem, where two models are trained against each other. This formulation provides high-quality results, but is very sensitive to hyperparameters and prone to diverging (illustrated in Figure 1b).

In this paper we minimize the sum of the forward and reverse KL divergence, also known as the (symmetric) Jeffreys divergence (Jeffreys, 1998). To accomplish this, we approximate the reverse KL divergence using a proxy model that simulates the true data distribution. We then use constrained optimization to formulate the joint problem of minimizing the Jeffreys divergence and fitting the proxy model (illustrated in Figure 1c). However, errors in the proxy model could hinder the approximation of the Jeffreys divergence and in extreme cases, render the constrained problem infeasible. We overcome this issue by using an *adaptive symmetrization* that dynamically adjusts the priority

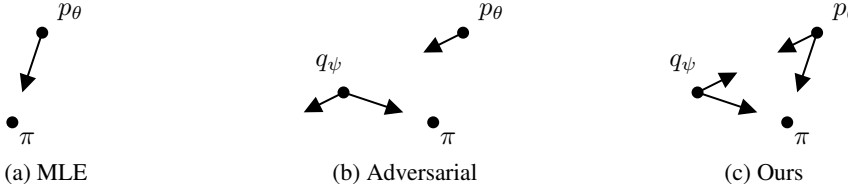

(a) MLE        (b) Adversarial       (c) Ours

Figure 1: Illustration of different training dynamics, where the goal is to train $q_\psi$ and $p_\theta$ to match the data distribution $\pi$. Each point represents a probability distribution in conceptual distribution space, and the arrows indicate the direction of gradient descent (GD). (a) In maximum likelihood estimation (MLE), GD minimizes the divergence $D_{\mathrm{KL}}(\pi \| p_\theta)$. In practice, GD minimizes a finite sample estimation of the KL, rather than the true KL. (b) GD of generative adversarial networks (GAN) pulls the generator $p_\theta$ towards the discriminator $q_\psi$ and the discriminator towards $\pi$, but it also pushes $q_\psi$ away from $p_\theta$. This repelling dynamic makes GAN training unstable. (c) In our framework, GD pulls $p_\theta$ and $q_\psi$ towards each other, while also pulling both towards $\pi$. The mutual attraction stabilizes training.

between optimizing the forward or reverse KL divergence, as well as fitting the proxy model. We develop a practical algorithm based on (non-convex) duality theory and the synergetic combination of NFs and EBMs. We compare our method with NF and Wasserstein GAN and find that our method is stable, robust and more accurate on a variety of problems, such as density estimation, image generation, and simulation-based inference (SBI).

To summarize, our contributions are: (1) We develop a practical method for optimizing the Jeffreys divergence, a symmetric version of the KL divergence, using constrained optimization and a proxy model; (2) We propose an adaptively symmetrized alternative to the Jeffreys divergence that enables different balances between the forward and reverse KL divergences. These balances are set automatically to trade off between optimizing each of these metrics and fitting the proxy model; (3) We develop a gradient descent-ascent algorithm to tackle this problem and simultaneously train an NF and an EBM. This joint training leverages the complementary strength of these models, mitigating their weaknesses,

## 2 PROBLEM FORMULATION

This paper seeks to determine the distribution $p_\theta \in \mathcal{H}$, parameterized by $\theta \in \mathbb{R}^k$, that is *closest* to an unknown probability distribution $\pi$ over $\mathbb{R}^m$. We assess this closeness using a set of $N$ independently and identically distributed (i.i.d.) observations $x_i$ from $\pi$. We assume all distributions are absolutely continuous with respect to Lebesgue measure and use the same for a distribution $p_\theta$ and its density $p_\theta(x)$. To formalize this problem, we next define (i) the hypothesis class $\mathcal{H}$ and (ii) a notion of distance.

**Hypothesis classes** There are as many ways to define $\mathcal{H}$ as there are ways of defining a probability distribution. A traditional choice from variational Bayesian inference considers a set of parametrized distributions, e.g., a Gaussian mixture model (GMM) (McLachlan & Peel, 2000). A more flexible approach used by *normalizing flows* (NFs) consists of letting $p_\theta \in \mathcal{H}$ be the pushforward of a tractable distribution $p_0$ by an invertible function $g_\theta : \mathbb{R}^m \to \mathbb{R}^m$, namely,

$$p_\theta(x) = p_0\big(g_\theta^{-1}(x)\big)\big|\det Dg_\theta^{-1}(x)\big| \tag{1}$$

where $Dg_\theta^{-1}$ denotes the Jacobian of the inverse of $g_\theta$. While $g_\theta$ is often a neural network (NN), its architecture has to be carefully designed to ensure that the its inverse and Jacobian are easy to compute (Dinh et al., 2017). This restriction reduces the flexibility of $\mathcal{H}$, but has the advantage of allowing NFs to efficiently sample from the distribution $p_\theta$ *as well as* evaluate its density.

A less restricted class is that of *energy-based models* (EBMs) that are based on unnormalized representations of distributions. Explicitly, EBMs are parametrized by an arbitrary function $f_\theta : \mathbb{R}^m \to \mathbb{R}$ that imply the density

$$p_\theta(x) = \frac{e^{f_\theta(x)}}{\zeta_\theta}, \quad \text{with } \zeta_\theta = \int e^{f_\theta(y)}\mathrm{d}y. \tag{2}$$

Since there are no restrictions on $f_\theta$, EBMs offer more powerful distribution models, though it is typically impossible to evaluate their density $p_\theta$ since estimating the partition function $\zeta_\theta$ in (2) is difficult even in moderate dimensions (Kloeckner, 2012). Nevertheless, it is possible to obtain samples directly from $f_\theta$ using Monte Carlo Markov Chain (MCMC) methods such as the Langevin Monte Carlo (LMC) algorithm (Parisi, 1981; Grenander & Miller, 1994). Further details about NF and EBM are provided in Appendices A.1 and A.2.

**Statistical distances** There is a plethora of ways to measure the distance between $p_\theta$ and $\pi$, from integral probability metrics (e.g., the Wasserstein distance) to $f$-divergences (Rényi, 1961; Ali & Silvey, 1966; Csiszár, 1967). While they all find particular use cases, the Kullback-Leibler (KL) divergence $D_{\mathrm{KL}}$ (Kullback & Leibler, 1951) remains a popular choice as it can be optimized as

$$\underset{p \in \mathcal{H}}{\mathrm{minimize}} \; D_{\mathrm{KL}}(\pi \parallel p) \triangleq \mathbb{E}_{x \sim \pi}\left[\log\left(\frac{\pi(x)}{p(x)}\right)\right] = -H(\pi) - \mathbb{E}_{x \sim \pi}[\log p(x)]. \tag{P-KL}$$

Note that the entropy $H(\pi) = \mathbb{E}_{x \sim \pi}[-\log \pi(x)]$, while unknown, is independent of $p$. Hence, (P-KL) is equivalent to a statistical learning problem whose solution can be approximated combining empirical risk minimization (ERM) and the parameterizations in (1) or (2). Explicitly,

$$\underset{\theta \in \mathbb{R}^k}{\mathrm{minimize}} \, \mathrm{NLL}\,(p_\theta) \triangleq -\frac{1}{N} \sum_{i=1}^{N} \log p_\theta(x_i). \tag{P-NLL}$$

Hence, (P-KL) is equivalent to minimizing the negative log-likelihood (NLL), i.e., solving a maximum likelihood estimation (MLE) problem (Huber et al., 1967).

Though practical, the KL divergence is not symmetric. Indeed, minimizing the *forward* KL divergence $D_{\mathrm{KL}}(\pi \parallel p_\theta)$ as in (P-KL), leads to solutions that cover the support of $\pi$ at the cost of overlooking sharp modes. In contrast, the *reverse* KL divergence $D_{\mathrm{KL}}(p_\theta \parallel \pi)$ promotes a mode-seeking behavior that may ignore regions of $\pi$ with significant mass (Murphy, 2012). One way to overcome this asymmetry is combining forward and reverse divergences as in

$$\underset{\theta \in \mathbb{R}^p}{\mathrm{minimize}} \; D_{\mathrm{J}}(p, \pi) \triangleq D_{\mathrm{KL}}(\pi \parallel p_\theta) + D_{\mathrm{KL}}(p_\theta \parallel \pi). \tag{P-JD}$$

The metric $D_{\mathrm{J}}$ is referred to as the *Jeffreys divergence* (JD) (Jeffreys, 1998; Kullback & Leibler, 1951). Yet, in contrast to its forward counterpart, the reverse KL divergence $D_{\mathrm{KL}}(p_\theta \parallel \pi)$ in (P-JD) is difficult to estimate or optimize based only on samples, as it requires access to the unknown density $\pi$.

Generative adversarial networks (GANs) (Goodfellow et al., 2014; Arjovsky et al., 2017) avoid this challenge by representing a symmetric divergence, such as the Jensen-Shannon divergence, as a variational problem (see Appendix A.3). This formulation results in a brittle min-max problem susceptible to diverging and an extreme form of mode-seeking known as *mode collapse* (Arjovsky & Bottou, 2017; Farnia & Ozdaglar, 2020). Instead, we next tackle the issues facing (P-JD) using (i) a proxy model and (ii) constrained optimization.

## 3 MINIMIZING THE JEFFREYS DIVERGENCE

In this section, we tackle the symmetrized (P-JD) by approximating the reverse KL using a proxy model and constrained optimization (Section 3.1). We then discuss the practical benefits of adapting the level of symmetrization throughout training, which we enable by rewriting (P-JD) as a feasibility problem with optimizable constraint specifications (Section 3.2).

### 3.1 APPROXIMATING THE REVERSE KL

A key roadblock in solving (P-JD) is the fact that the reverse KL divergence $D_{\mathrm{KL}}(p_\theta \parallel \pi)$ directly depends on the unknown data density $\pi$. Hence, the reverse KL can only be computed after estimating $\pi$ from its samples using, e.g., kernel density estimation or k-nearest neighbors (Silverman, 2018). Though effective for estimation, these methods come with bias-variance trade-offs that are difficult to balance, especially as $p_\theta$ evolves during training.

To overcome this limitation, we incorporate this data density estimation step directly in (P-JD). Explicitly, consider a proxy distribution model $q_\psi \in \mathcal{H}_\psi$ parametrized by $\psi \in \mathbb{R}^k$ (we assume that $\theta$ and $\psi$ have the same dimensions without loss of generality). As long as $q_\psi \approx \pi$, we can use it to train $p_\theta$ by computing the reverse KL divergence in the objective of (P-JD) as $D_{\mathrm{KL}}(p_\theta \parallel q_\psi)$. Using constrained optimization, we can simultaneously minimize the Jeffreys divergence, ensure that $q_\psi$ fits the data distribution. Beyond that, we can impose additional constraints, e.g., to regularize the models. Explicitly, we replace (P-JD) by

$$\begin{aligned} \underset{\theta, \psi \in \mathbb{R}^k}{\text{minimize}} \quad & D_{\mathrm{KL}}(\pi \parallel p_\theta) + D_{\mathrm{KL}}(p_\theta \parallel q_\psi) \\ \text{subject to} \quad & D_{\mathrm{KL}}(\pi \parallel q_\psi) \le \epsilon, \quad h(p_\theta, q_\psi) \le c, \end{aligned} \tag{PI}$$

where $\epsilon \ge 0$ determines the accuracy of the reverse KL approximation and $h, c$ define (optional) task-dependent constraints. Notice that the proxy model $q_\psi$ is simultaneously trained to fit $\pi$ (constraint) as well as used to evaluate the reverse KL of $p_\theta$ (objective). In general, $\mathcal{H}_\psi$ need not be the same as $\mathcal{H}$ and it might in fact be beneficial to choose different model types (see Section 4.2).

## 3.2 Adaptive symmetrization

Depending on the expressiveness of the proxy model $q_\psi$, it may be impossible to choose a small enough value for $\epsilon$ to ensure that it is close enough to $\pi$. Even when it is possible, unless we ensure that $q_\psi$ is close enough to $\pi$ throughout training, there could be phases during which the reverse KL approximation is inaccurate. In such situations, we may want to put more emphasis on the forward divergence in the objective of (PI), since it is computed directly from data. In other words, we may want to (temporarily) break the symmetry of the Jeffreys divergence while training $q_\psi$ to fit $\pi$. As $q_\psi$ starts to better reflect the data distribution, we can go back to balancing the symmetrization. Dynamically changing the focus between forward and reverse KL divergences steers training between mode-covering and mode-seeking behaviors.

Naturally, it would be impractical to manually adjust these trade-offs throughout training, especially since they depend on the model classes $\mathcal{H}, \mathcal{H}_\psi$ and the data. We therefore rely on the resilient optimization framework proposed in Chamon et al. (2020a;b); Hounie et al. (2023). This approach adjusts constraint specifications such as $\epsilon$ in (PI) depending on how difficult it is to fulfill. Accordingly, a constraint that is difficult (easy) to satisfy will have its specification increased (decreased). In its simplest form, this behavior can be achieved by writing (PI) as a feasibility problem and incorporating the constraint specifications in the objective as in

$$\begin{aligned} \underset{\substack{\theta, \psi \in \mathbb{R}^k \\ \epsilon_{\mathrm{fw}}, \epsilon_{\mathrm{rv}}, \epsilon_{\mathrm{prx}} \ge 0}}{\text{minimize}} \quad & \epsilon_{\mathrm{fw}}^2 + \epsilon_{\mathrm{rv}}^2 + \epsilon_{\mathrm{prx}}^2 \\ \text{subject to} \quad & D_{\mathrm{KL}}(\pi \parallel p_\theta) \le \epsilon_{\mathrm{fw}}, \quad D_{\mathrm{KL}}(p_\theta \parallel q_\psi) \le \epsilon_{\mathrm{rv}} \\ & D_{\mathrm{KL}}(\pi \parallel q_\psi) \le \epsilon_{\mathrm{prx}}, \quad h(p_\theta, q_\psi) \le c. \end{aligned} \tag{P-DYN}$$

For simplicity, we keep the additional requirements specification $c$ fixed. This formulation is equivalent, under mild conditions, to equalizing the relative sensitivity of the constraints (Chamon et al., 2020a;b; Hounie et al., 2023). In the next section, we develop a practical algorithm to tackle (P-DYN) using NFs and EBMs.

## 4 Algorithm development

Solving (P-DYN) is challenging due to its dependence on the unknown data distribution $\pi$. Even if we used the empirical approximation in (P-NLL), it would remain a non-convex constrained optimization problem for typical choices of $p_\theta, q_\psi$, including NFs and EBMs. It therefore lacks *strong duality* (Bertsekas, 2009; Bonnans & Shapiro, 2013). Although it might be appealing to use the penalty formulation, where each term is weighted by a fixed weight $w$ as

$$\underset{\theta, \psi \in \mathbb{R}^k}{\text{minimize}} \quad w_{\mathrm{fw}} D_{\mathrm{KL}}(\pi \parallel p_\theta) + w_{\mathrm{rv}} D_{\mathrm{KL}}(p_\theta \parallel q_\psi) + w_{\mathrm{prx}} D_{\mathrm{KL}}(\pi \parallel q_\psi) \tag{P-W}$$

for weights $w_{\mathrm{fw}}, w_{\mathrm{rv}}, w_{\mathrm{prx}} > 0$, it is not equivalent to solving (P-DYN). In fact, (P-W) leads to worse results in practice (see Section 5.1). To overcome this challenge, we use non-convex duality theory in Section 4.1. We then derive a practical algorithm by using NFs and EBMs (Section 4.2).

### 4.1 THE EMPIRICAL DUAL PROBLEM

We formulate the empirical version of (P-DYN) with the empirical mean over the data samples and incorporate the constant entropy of the data distributions in the constraint specifications $\epsilon$. Explicitly,

$$
P^\star \triangleq \min_{\substack{\theta, \psi \in \mathbb{R}^k \\ \epsilon \geq 0}} \quad \epsilon_{\text{fw}}^2 + \epsilon_{\text{rv}}^2 + \epsilon_{\text{prx}}^2
$$

$$
\text{subject to} \quad -\frac{1}{N} \sum_{i=1}^N \log p_\theta(x_i) \leq \epsilon_{\text{fw}}, \quad -\frac{1}{N} \sum_{i=1}^N \log q_\psi(x_i) \leq \epsilon_{\text{prx}} \qquad (\hat{\text{P}}\text{-DYN})
$$

$$
D_{\text{KL}}(p_\theta \parallel q_\psi) \leq \epsilon_{\text{rv}}, \qquad h(p_\theta, q_\psi) \leq c,
$$

where we collect $\epsilon_{\text{fw}}, \epsilon_{\text{rv}}, \epsilon_{\text{prx}}$ in the vector $\epsilon$ and write $\epsilon \geq 0$ to denote that it belongs to the non-negative orthant. We then define the dual problem of ($\hat{\text{P}}$-DYN) as

$$
D^\star = \max_{\boldsymbol{\lambda} \geq 0} \quad \min_{\theta, \psi \in \mathbb{R}^k, \epsilon \geq 0} \mathcal{L}(\theta, \psi, \epsilon, \boldsymbol{\lambda}) \qquad (\hat{\text{D}}\text{-DYN})
$$

for the Lagrangian

$$
\mathcal{L}(\theta, \psi, \epsilon, \boldsymbol{\lambda}) = \epsilon_{\text{fw}}^2 + \epsilon_{\text{rv}}^2 + \epsilon_{\text{prx}}^2 + \lambda_{\text{fw}} \left[ -\frac{1}{N} \sum_{i=1}^N \log p_\theta(x_i) - \epsilon_{\text{fw}} \right]
$$

$$
+ \lambda_{\text{rv}} \left[ D_{\text{KL}}(p_\theta \parallel q_\psi) - \epsilon_{\text{rv}} \right] + \lambda_{\text{prx}} \left[ -\frac{1}{N} \sum_{i=1}^N \log q_\psi(x_i) - \epsilon_{\text{prx}} \right] + \lambda_h \left[ h(p_\theta, q_\psi) - c \right], \quad (3)
$$

where the dual variables $\lambda_{\text{fw}}, \lambda_{\text{rv}}, \lambda_{\text{prx}}, \lambda_h$ are collected in the vector $\boldsymbol{\lambda}$. Observe from (3) that the objective of the dual problem ($\hat{\text{D}}$-DYN) is reminiscent of the penalty formulation in (P-W). In fact, they are essentially equivalent for $\epsilon = 0$ and $\lambda_h = 0$. The crucial distinction is that the dual variables $\boldsymbol{\lambda}$ are optimization parameters of ($\hat{\text{D}}$-DYN), whereas the weights $w$ in (P-W) are fixed. This distinction allows us to show that ($\hat{\text{D}}$-DYN) approximates ($\hat{\text{P}}$-DYN) despite non-convexity:

**Theorem 1.** *Let $p_\theta, q_\psi > 0$ and $h$ be a convex and Lipschitz continuous function. Suppose there exists $\nu \geq 0$ such that for each $p \in \bar{\mathcal{H}}$, the closed convex hull of $\mathcal{H}$, there exists $p_\theta \in \mathcal{H}$ such that $\|p_\theta - p\|_{TV} \leq \nu$. Suppose the same holds for $\mathcal{H}_\psi$. Then, there exists a finite constant $B$ such that*

$$
0 \leq P^\star - D^\star \leq B\nu. \qquad (4)
$$

*Proof.* The result follows from Chamon et al. (2023, Prop. III.3, Lemma III.1) by noticing that (i) for $p_\theta, q_\psi > 0$, the losses in ($\hat{\text{P}}$-DYN) are Lipschitz continuous (Assumption 1 in Prop. III.3), (ii) $\mathcal{H}$ and $\mathcal{H}_\psi$ satisfy Assumption 3 by hypothesis, and (iii) Assumption 4 is satisfied trivially since $\epsilon$ is an optimization variable. Assumption 2 is not needed for either result. $\square$

Theorem 1 enables us to (approximately) solve the constrained problem ($\hat{\text{P}}$-DYN) by means of its unconstrained dual problem ($\hat{\text{D}}$-DYN). The latter can be tackled using gradient descent-ascent (GDA) techniques common in primal-dual methods (Boyd & Vandenberghe, 2004; Chamon & Ribeiro, 2020; Chamon et al., 2023; Elenter et al., 2024). They involve taking (projected) gradient descent steps for

---

**Algorithm 1** Gradient descent-ascent

**Require:** Learning rates $\alpha_\psi, \alpha_\theta, \alpha_\lambda, \alpha_\epsilon$
1: $\psi \leftarrow \psi - \alpha_\psi \nabla_\psi \mathcal{L}$
2: $\theta \leftarrow \theta - \alpha_\theta \nabla_\theta \mathcal{L}$
3: $\epsilon \leftarrow \max\{0, \epsilon - \alpha_\epsilon \nabla_\epsilon \mathcal{L}\}$
4: $\boldsymbol{\lambda} \leftarrow \max\{0, \boldsymbol{\lambda} + \alpha_\lambda \nabla_{\boldsymbol{\lambda}} \mathcal{L}\}$

---

the primal optimization variables $\theta, \psi, \epsilon$ and (projected) gradient ascent steps for the dual variables $\boldsymbol{\lambda}$ (see Algorithm 1). Theorem 1 also shows that the approximation error can be made arbitrarily small by sufficiently rich models for $p_\theta$ and $q_\psi$. Their choice also affects the implementation of Algorithm 1, as we show next.

### 4.2 CHOOSING THE MAIN AND PROXY MODELS

The dual problem ($\hat{\text{D}}$-DYN) jointly optimizes two models, similarly to GANs. Yet the GAN objective (Appendix A.3) permits models without explicit densities, while ($\hat{\text{D}}$-DYN) enforces parameterized distributions. Therefore, natural choices for $p_\theta$ and $q_\psi$ are either NFs or EBMs. NFs directly

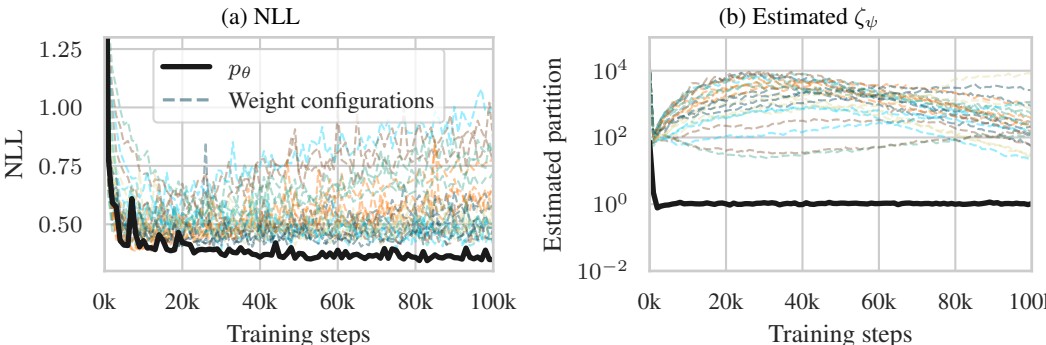

Figure 2: Solving the dual problem (solid black line) achieves better results than 25 weight configurations of the weighted problem Equation (P-W) (dashed colored lines) over a synthetic Gaussian mixture dataset. (a) The $p_\theta$ of dual problem has better and more stable NLL than any configuration of the weighted problem. (b) The partition function of the EBM of the dual problem is stable at unity, while the weighted problem may reach large unstable values.

parametrize a probability distribution (Equation (1)) and are easy to evaluate and optimize, making them the ideal choice. However, NF's restricted architecture may prevent them from perfectly capturing $\pi$, introducing an irreducible error to the reverse KL approximation and empirical dual problem (D̂-DYN). This hindrance suggests that EBMs are a better choice, as they have virtually unlimited expressivity.

We propose to use NF for $p_\theta$ and EBM for $q_\psi$ to exploit their complementary nature to overcome their limitations. The NF makes the log-likelihood of $p_\theta$ straightforward to evaluate, while the EBM's versatility reduces the errors incurred from using a proxy model. This symbiotic relationship between NFs and EBMs has been observed in several prior works (see Appendix A.5). At the same time, since EBMs are unnormalized (see Equation (2)), evaluating and optimizing log-likelihoods and KL divergences involving $q_\psi$ require obtaining (a potentially large number of) samples from its distribution, which calls for time-consuming MCMC methods. However, it is straightforward to both obtain samples from the NF $p_\theta$ as well as evaluate their density. The partition function of the EBM $q_\psi$ can therefore be estimated using importance sampling as in

$$\zeta_\psi = \int e^{f_\psi(y)} \frac{p_\theta(y)}{p_\theta(y)} dy = \mathbb{E}_{y \sim p_\theta} \left[ e^{f_\psi(y) - \log p_\theta(y)} \right]. \tag{5}$$

This replaces the slow EBM sampling by the faster NF. Furthermore, notice that $\epsilon_{rv}$ in (P̂-DYN) has the effect of keeping $p_\theta$ and $q_\psi$ close to each other, reducing the variance of the importance sampling estimate in (5) (Kloek & van Dijk, 1978; Chatterjee & Diaconis, 2018; Sanz-Alonso, 2018). Still, we can improve numerical stability by ensuring that $\zeta_\psi$ remains close to one, i.e., that $q_\psi$ is almost normalized. This can be done in (P̂-DYN) by using $h(q_\psi) = [\zeta_\psi - 1, \ 1 - \zeta_\psi]^\top$.

Using these observations, we can obtain a practical version of Algorithm 1 to tackle (D̂-DYN) and, consequently, the adaptively symmetrized KL divergence optimization problem (P̂-DYN) (see Theorem 1). We present the full algorithm in Appendix B.

## 5 EXPERIMENTAL RESULTS

### 5.1 VALIDATION AND COMPARISON

We begin by illustrating the advantages of our method by comparing our approach to other baselines. First, we compare our method to the penalty approach (P-W) with fixed weights. To that end, we train both formulations on a synthetic 2D dataset of 800 points drawn from a GMM with 40 components (Midgley et al., 2023) (see Appendix D.1 for the technical details of this experiment). We measure the NLL and partition function over a test dataset of 10k samples. Since we approximate the partition function of $q_\psi$, the exact NLL is unknown, and the values we report are estimations. To

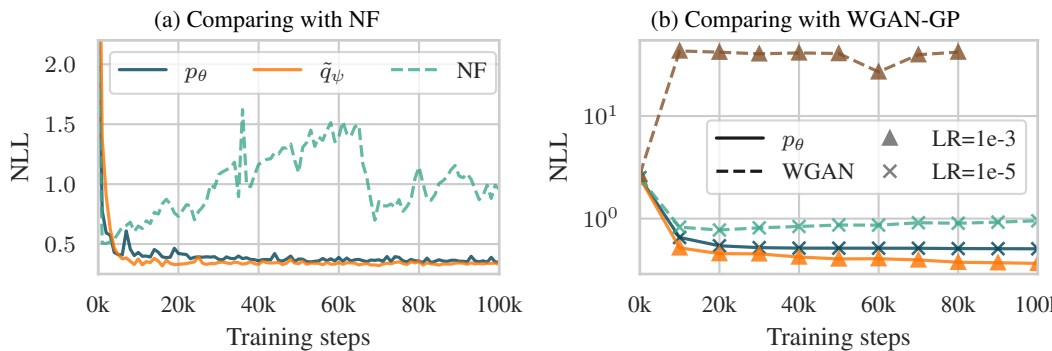

Figure 3: Our dual is more stable than NF and WGAN and outperforms them on a synthetic 2D GMM dataset. (a) The dual problem (solid lines) has a stable KL divergence lower than NF (dashed line). NF's KL is increasing since it overfits to a finite distribution rather than the test GMM distribution. (b) WGANs (dashed lines) is unstable when increasing the learning rate, while our method (solid lines) is stable. Triangles represent high learning rate, and X represents low learning rate.

make this fact explicit, we denote the quasi-normalized density of $q_\psi$ as $\tilde{q}_\psi$. We compare 25 different weight configurations for (P-W) against a single instance of ($\hat{\text{D}}$-DYN) in Figure 2a. The adaptive weights of the dual problem attain lower and more stable NLL than any of the weight combinations we tried. Additionally, Figure 2b shows the numerical stability of ($\hat{\text{D}}$-DYN) compared to (P-W) due to its constraint on the partition function $\zeta_\psi$ of the EBM. This constraint effectively controls the value of $\zeta_\psi$, which in contrast can become large when solving (P-W). This instability issue can also be addressed through other means, such as regularization or projected gradient methods.

We also compare our adaptive symmetrization approach with the traditional forward KL divergence used to train NFs. To do so, we use the same dataset, architecture and learning rate as $p_\theta$. While the pure NF attains low NLL initially, the asymmetry of the forward KL divergence overfits the training set and leads to higher NLL on the actual test distribution $\pi$. However, both $p_\theta$ and $q_\psi$ remain stable at a lower NLL, confirming the advantage of the symmetrized divergence.

Finally, we compare with Wasserstein GAN with gradient penalty (WGAN-GP) (Gulrajani et al., 2017), which minimizes the symmetric Wasserstein distance. To compare with our formulation, we use the same architecture and learning rate (LR) for $p_\theta$ and the WGAN's generator and the same architecture and LR for $q_\psi$ and the WGAN's critic. This is a fair comparison since the critic (or discriminator) in GANs are implicit EBMs (Che et al., 2020; Ben-Dov et al., 2024). We then tested two learning rates for both $p_\theta$ and the generator and present the results in Figure 3b. Note that the LR of $q_\psi$ was reduced by an order of magnitude from the previous experiments to keep the WGAN training stable. As expected, WGAN is stable only at low learning rates due to the opposing objectives of the generator and the critic. Since our approach is collaborative rather than adversarial, it remains stable over a wider range of LRs and reaches lower NLL values.

## 5.2 DENSITY ESTIMATION

We test the density estimation capabilities of our method on five common 2D datasets: Gaussian mixture ring, Gaussian mixture grid, two moons, spiral, and concentric circles. Some of these distribution have low entropy and result in negative NLL, which is not considered by the resilient approach in (P-DYN). To compensate for that, we use a modified formulation that accommodates negative constraints (see Appendix B.7). Figure 4 presents two of these datasets, while Appendix D.2 presents the rest and details the training configurations. In all the datasets, our method learn a density with lower and more consistent NLL than a pure NF (Figures 4a and 4c). Moreover, the qualitative density maps in Figures 4b and 4d show that both models perfectly capture the shape and all modes.

## 5.3 SAMPLING

Next, we illustrate the sampling quality of $p_\theta$ by learning to sample from the image dataset CelebA (Liu et al., 2015). The main problem when training on a high-dimensional space is that

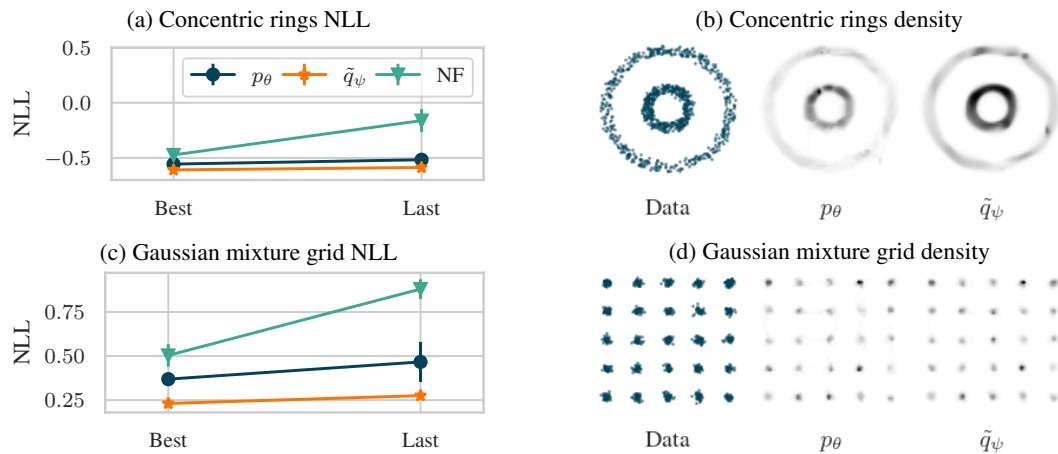

Figure 4: Our framework is able to accurately learn the density of various 2D datasets. The left column (a,c) present the lowest NLL each model achieved during training ("Best") and the NLL at the end of training ("Last"). The error bars represent the standard deviation over 5 seeded runs, demonstrating that our method outperforms NF in terms of values and consistency. The right column (b,d) qualitatively displays the finite dataset (left) and the learned density values from of $p_\theta$ and the quasi-normalized $\tilde{p}_\psi$ (middle and right respectively). The qualitative density maps show that both models perfectly capture the shape and all modes.

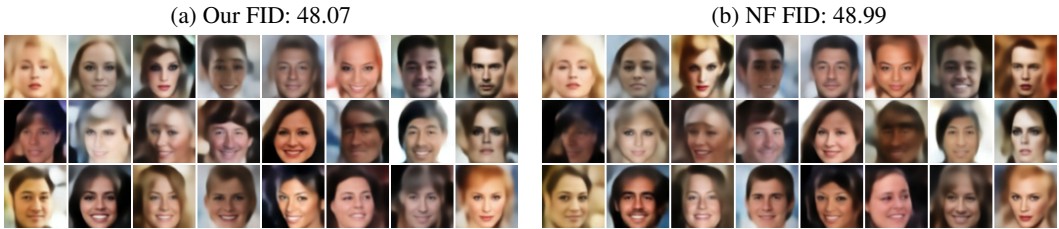

Figure 5: Our $p_\theta$ learns a comparable 100D encoded space of CelebA as NF. (a) reports the FID and qualitative samples generated from our $p_\theta$, and (b) reports the same for NF. We used the same seed to generate the sample, which explains the similarity between the images.

the gradient of $\psi$ may explode due to the exponential in $\nabla_\psi \mathcal{L}$ (see Appendix B). We mitigate this issue by taking three steps. First, we train an autoencoder on the training dataset and learn a 100D encodings that serve as the training data for our models. Second, we add a temperature parameter to the EBM that scales the density values to the correct order of magnitude. Third, before training we warm start the EBM by training it to have a similar density as the NF. Appendix B.8 details the implementation of these steps and Appendix D.3 the training details. To compare between our method and a pure NF, we use the Fréchet inception distance (FID) (Heusel et al., 2017), a common metric to measure the generation quality of a model. Figure 5 displays the decoded samples from $p_\theta$ and an NF and their FID values. Our model gets comparable FID to NF.

## 5.4 SIMULATION BASED INFERENCE

Here we utilize our dual formulation to to learn conditional distributions for simulations-based inference (SBI). Many scientific fields attempt to determine the parameters of a mechanism $\beta$ given observations of its outcome $u$. However, even with perfect knowledge of the mechanism and the ability to predict $\pi(u|\beta)$, the inverse — inferring the parameters given the outcome $\pi(\beta|u)$ — is challenging when the mechanism is stochastic. For example, despite having a perfect model to predict gravitational waves from a known source, it is harder to infer the sources given a wave (Dax et al., 2021; Wildberger et al., 2023). SBI provides solutions to amortized inference by training a model on simulated instances (Cranmer et al., 2020). A simple method to perform this inference is neural posterior estimation (NPE) (Papamakarios & Murray, 2016; Lueckmann et al., 2017; Radev et al., 2022). This requires to define a prior distribution $p_\beta$ over the parameters, sample $N$ instance

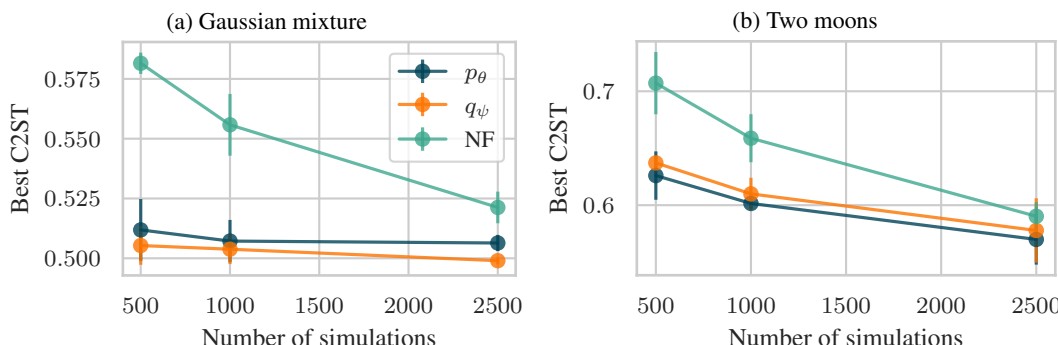

Figure 6: Our method requires less simulations than NF to achieve C2ST scores closer to 0.5 for common SBI benchmarks. For each method, over 5 different seeds, we compute the best C2ST over the run and report the mean and standard deviation as the error bars.

of $\beta$, and simulate $N$ outcomes $\pi(u|\beta)$. The conditional inference network then trains on the pairs $(\beta, u)$ to learn the posterior $\pi(\beta|u)$. Prior work shows that maximizing the entropy of the learned distribution improves the chances of recovering the underlying distribution (Vetter et al., 2024). While prior works add an entropy regularization term, note that the Lagrangian in Equation (3) naturally contains the entropy of $p_\theta$ due to the reverse KL.

In our experiments, we use the conditional version of ($\hat{\text{P}}$-DYN), which we develop in Appendix B.6. We trained our method on the common SBI benchmarks two moons (Greenberg et al., 2019), and Gaussian mixture (Sisson et al., 2007). As common in the SBI literature (Ramesh et al., 2022; Vetter et al., 2024), we measure the effect of the size of the dataset (number of simulations) on the quality of the learned posterior. See Appendix D.4 for the full technical details. For evaluation, we used the classifier-two-sample testing (C2ST). The C2ST score is defined as the accuracy of a classifier trained to distinguish between samples from the ground truth posterior and samples from the learned posterior. The optimal value is 0.5, indicating that the generated posterior is indistinguishable from the ground truth posteriors. To train the classifier, we created a data set by sampling 5000 points from both the ground truth posterior and our models, conditioned on the same pre-determined $u_0$ and labeled them accordingly. We run the experiment 5 times with different seeds and report the results in Figure 6. Our method generates higher quality posterior samples than a pure NF when trained on the same number of samples.

# 6 CONCLUSION

Learning a distribution from finite samples commonly requires minimizing the asymmetric KL divergence or a symmetric divergence via an unstable min-max objective. In this paper we minimize the symmetric Jeffreys divergence by having two models collaborate rather than being adverse. The additional model's task is to mimic the data distribution, which we use to estimate the reverse KL divergence. We use a constrained optimization problem to jointly train the two models by solving the unconstrained dual problem. We demonstrate on various tasks that our method is stable, and requires less data points to arrive at the same accuracy as other methods. An additional benefit of our method is that while proxy model (EBM) is used to assist the training of the primary model (NF), it is also itself trained on the data and can be used to potentially obtain higher quality samples.

We find this formulation to have exciting future research directions. For example, it would be interesting to extend this method to work in high-dimensional settings by estimating the probability of generated samples (Ben-Dov et al., 2024) or by using rectangular flows (Caterini et al., 2021). Also, while estimating the partition $\zeta_\psi$ using samples from $p_\theta$ is relatively efficient (Appendix E), high dimensions may require streamlining this process by adaptively changing the number of samples and the frequency of estimation. A theoretical direction involves using other models and divergences in (P-DYN), such as score-based models with the Fisher divergence. One form of the Fisher divergence avoids the score of $\pi$ by computing the Hessian (Hyvärinen, 2005). With our formulation, rather than the expensive computations of the Hessian, we can use the score of a tractable proxy model. Future work also includes training the models with additional constraints for specific tasks, such as fairness or privacy.

## REPRODUCIBILITY STATEMENT

We provide the technical details to reproduce our experiments in the appendix. In Appendix B we detail our algorithm and models and in Appendix D we specify the specific of each experiment.

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

## 7 APPENDIX

## A RELATED WORK

### A.1 ENERGY-BASED MODELS

Any function $f_\psi : \mathbb{R}^m \to \mathbb{R}$ has a corresponding probability distribution

$$q_\psi(x) = \frac{e^{f_\psi(x)}}{\zeta_\psi}, \quad \text{with} \quad \zeta_\psi = \int e^{f_\psi(y)} \mathrm{d}y. \tag{6}$$

The exponent keeps the output non-negative and the partition function $\zeta_\psi$ ensures that the distribution integrates to one. Essentially, Equation (2) is the definition of a Boltzmann distribution with an energy function $-f_\psi$, consequently naming this class of functions as energy-based models (Teh et al., 2003; Jianwen Xie et al., 2016; Du & Mordatch, 2019).

The gradient of the NLL objective in Equation (P-KL) with $q_\psi$ from Equation (2) is

$$\begin{aligned}
\nabla_\theta \mathbb{E}_{x \sim \pi} \left[ -\log q_\psi(x) \right] &= \mathbb{E}_{x \sim \pi} \left[ -\nabla_\theta f_\psi(x) + \nabla_\theta \log \zeta_\psi \right] \\
&= \mathbb{E}_{x \sim \pi} \left[ -\nabla_\theta f_\psi(x) \right] + \nabla_\theta \log \zeta_\psi \\
&= \mathbb{E}_{x \sim \pi} \left[ -\nabla_\theta f_\psi(x) \right] + \frac{1}{\zeta_\psi} \nabla_\theta \int e^{f_\psi(y)} \mathrm{d}y \\
&= \mathbb{E}_{x \sim \pi} \left[ -\nabla_\theta f_\psi(x) \right] + \int \frac{1}{\zeta_\psi} e^{f_\psi(y)} \nabla_\theta f_\psi(y) \, \mathrm{d}y \\
&= \mathbb{E}_{x \sim \pi} \left[ -\nabla_\theta f_\psi(x) \right] + \int q_\psi(y) \nabla_\theta f_\psi(y) \, \mathrm{d}y \\
&= \mathbb{E}_{x \sim \pi} \left[ -\nabla_\theta f_\psi(x) \right] + \mathbb{E}_{y \sim q_\psi} \left[ \nabla_\theta f_\psi(y) \right]
\end{aligned} \tag{7}$$

The second term of the gradient requires sampling from the EBM itself (Du & Mordatch, 2019; Song & Kingma, 2021). Sampling from $q_\psi$ is not straightforward and typically involves running a Markov Chain Monte Carlo (MCMC) method, such as Langevin dynamics (Parisi, 1981; Grenander & Miller, 1994). Starting with $x_0$ from a simple prior distribution, the $k$-th Langevin step is

$$x_k = x_{k-1} + s \nabla_x \log q_\psi(x) + \sqrt{2s} z_{k-1}, \tag{8}$$

with $s$ being the step-size and $z \sim \mathcal{N}(\mathbf{0}, I)$. EBMs are a powerful class of functions due to their unrestricted architecture, but the Langevin sampling makes their training slow and requires more hyperparameter tuning.

### A.2 NORMALIZING FLOWS

Unlike EBMs, which directly define the unnormalized density, NFs define an invertible mapping $f_\theta : \mathbb{R}^m \to \mathbb{R}^m$, parameterized by $\theta$, from a simple base distribution $p_0$ to the data distribution (Tabak & Vanden-Eijnden, 2010; Tabak & Turner, 2013; Danilo Rezende & Shakir Mohamed, 2015; Papamakarios et al., 2021). Training NF involves minimizing the NLL from Equation (P-KL) through the change-of-variables law of probabilities

$$\log p_\theta(x) = \log p_0\left(f_\theta^{-1}(x)\right) + \log \left| \det \frac{\mathrm{d} f_\theta^{-1}(x)}{\mathrm{d}x} \right|, \tag{9}$$

where $\frac{\mathrm{d} f_\theta^{-1}(x)}{\mathrm{d}x}$ is the Jacobian of $f_\theta^{-1}$ at $x$. While computing the Jacobian of a multivariate function is computationally expensive, practical NF relies on architectures with tractable Jacobians (Dinh et al., 2017; Kingma & Dhariwal, 2018; Durkan et al., 2019). These Jacobians simplify the training of NF, and sampling is quick compared to EBMs. Yet, since the forward KL is minimized from a finite dataset, NF tend to overfit to the empirical distribution rather than the underlying distribution. This problem is usually addressed by adding random noise to the data (Uria et al., 2013; Ho et al., 2019). Sampling from NF is quicker than sampling from an EBM as it requires only sampling from the base distribution and applying the invertible mapping.

### A.3 Generative adversarial networks

When minimizing only the empirical forward $D_{\mathrm{KL}}$, the model learns to give high probability to the training samples, but may assign high probability to regions where $\pi$ is low, i.e., have low precision. This is due to the asymmetry of the KL divergence $D_{\mathrm{KL}}\left(p \parallel q\right) \neq D_{\mathrm{KL}}\left(q \parallel p\right)$, which relies on sampling from only one of the distributions. Minimizing a symmetric divergence may avoid this pitfall, but generally requires access to the unknown $\pi$. However, some symmetric divergences can be expressed as a variational representation without direct access to $\pi$. For example, minimizing any $f$-divergence, a divergence $D_f$ defined by a function $f$, can be formulated as the adversarial min-max objective (Nguyen et al., 2010; Nowozin et al., 2016)

$$\min_{\theta} D_f\left(\pi \parallel p_\theta\right) = \min_{\theta} \max_{\psi}\left(\mathbb{E}_{x \sim \pi}\left[g_\psi\left(x\right)\right] - \mathbb{E}_{x \sim p_\theta}\left[f^*\left(g_\psi\left(x\right)\right)\right]\right), \tag{10}$$

where $g_\psi$ is a parameterized function and $f^*$ is the convex conjugate of $f$. The most common applications of this formulation is the generative adversarial networks (GAN) (Goodfellow et al., 2014) minimizing the Jensen-Shannon divergence, and WGAN (Arjovsky et al., 2017) which minimizes the Wasserstein distance[1]. The advantages of adversarial training is the efficient sampling from the generator $p_\theta$, and that the discriminator $g_\psi$ can be used as an EBM (Che et al., 2020; Ben-Dov et al., 2024). Yet, GANs' adversarial minimax game between the two models may not always converge to a stable solution, may have vanishing gradients or suffer mode collapse (Arjovsky & Bottou, 2017; Mescheder et al., 2018; Farnia & Ozdaglar, 2020). In this paper we propose a non-adversarial objective to train a symmetrized divergence.

### A.4 Minimizing the forward and reverse KL

Prior attempts to explicitly minimize both the forward and reverse KL divergences include the $\alpha$-bridge (Zhao et al., 2020). This method leverages the definition of an $\alpha$-divergence (Cichocki & Amari, 2010) which converges to the forward and reverse KL in the limits $\alpha \to 0^+$ and $\alpha \to 1^-$ by training model in three steps. First, the algorithm minimizes the forward KL by maximizing the evidence lower bound (ELBO), similarly to variational autoencoders (VAE). Second, the algorithm increases $\alpha$ according to a fixed schedule and minimizes the corresponding $\alpha$-divergence. Third, the algorithm minimizes the reverse KL through adversarial training (as the $f$-divergence in Appendix A.3). Our method (Section 4) avoids the ELBO approximation and adversarial training by jointly optimizing both divergences in a single objective. Additionally, rather than a fixed schedule to balance between the two divergence, our method dynamically adjusts the weight of each direction during optimization.

### A.5 Jointly training EBM and NF

NF are fast samplers but have low expressiveness compared to EBM which are flexible, but requires a slow MCMC sampling process. This contrast encouraged research into jointly training these models so each model's strength covers for the other's weakness. Xie et al. (2022) propose to fit an NF to an EBM, which will then generate an initial starting point for the Langevin dynamics in the training of the EBM. This approach accelerates the training of the EBM but keeps the MCMC process, albeit shortening the mixing time. Other approaches remove the MCMC by installing the EBM and NF in a GAN setting (Grover et al., 2018; Gao et al., 2020; Ben-Dov et al., 2024). While these works avoid the slow EBM sampling, they inherit the instability of adversarial training.

---

[1]Wasserstein distance is not an $f$-divergence, but can also be expressed as a variational problem with a 1-Lipschitz constraint over $g_\psi$.

## B ALGORITHM

### B.1 PRACTICAL PROBLEM FORMULATION

In practice, we formulate Equation ($\hat{\text{P}}$-DYN) in our experiments by adding an additional constraint over the normalization of the EBM as

$$\operatorname*{minimize}_{\psi,\theta,\boldsymbol{\epsilon}\geq 0} \quad \epsilon_{\text{fw}}^2 + \epsilon_{\text{rv}}^2 + \epsilon_{\text{EBM}}^2$$

$$\text{subject to} \quad -\frac{1}{N}\sum_{i=1}^{N} \log p_\theta\left(x_i\right) \leq \epsilon_{\text{fw}}$$

$$\frac{1}{N}\sum_{i=1}^{N} \left[\log p_\theta\left(y_i\right) - \log f_\psi\left(y_i\right)\right] + \log \zeta_\psi \leq \epsilon_{\text{rv}} \tag{PII}$$

$$-\frac{1}{N}\sum_{i=1}^{N} \left[\log f_\psi\left(x_i\right)\right] + \log \zeta_\psi \leq \epsilon_{\text{EBM}}$$

$$1 \leq \zeta_\psi \leq 1 + \epsilon_\zeta,$$

where $x_i$ are the data points and $y_i$ are samples from $p_\theta$. The corresponding Lagrangian is

$$\mathcal{L}\left(\psi,\theta,\boldsymbol{\epsilon};\boldsymbol{\lambda}\right) = \frac{1}{N}\sum_{i=1}^{N} \left[-\lambda_{\text{fw}}\log p_\theta\left(x_i\right) - \lambda_{\text{prx}}\log f_\psi\left(x_i\right) + \lambda_{\text{rv}}\log p_\theta\left(y_i\right) - \lambda_{\text{rv}}\log f_\psi\left(y_i\right)\right]$$

$$+ \epsilon_{\text{fw}}^2 - \lambda_{\text{fw}}\epsilon_{\text{fw}} + \epsilon_{\text{rv}}^2 - \lambda_{\text{rv}}\epsilon_{\text{rv}} + \epsilon_{\text{EBM}}^2 - \lambda_{\text{prx}}\epsilon_{\text{EBM}}$$

$$+ \lambda_{\text{u}}\left(-1 - \epsilon_\zeta\right) + \lambda_\ell + \left(\lambda_{\text{u}} - \lambda_\ell\right)\zeta_\psi + \left(\lambda_{\text{prx}} + \lambda_{\text{rv}}\right)\log \zeta_\psi. \tag{11}$$

The partition function is empirically estimated as

$$\zeta_\psi \approx \frac{1}{M}\sum_{i=1}^{M} \left[e^{f_\psi(y_i) - \log p_\theta(y_i)}\right], \quad \text{with } y_i \sim p_\theta. \tag{12}$$

### B.2 CLOSED FORM SOLUTION FOR $\boldsymbol{\epsilon}$

Given $\boldsymbol{\lambda}$, equating the derivative $\nabla_{\boldsymbol{\epsilon}}\mathcal{L}$ to zero, results in a closed form solution

$$\begin{aligned} 0 &= \nabla_{\boldsymbol{\epsilon}}\mathcal{L} \\ &= \nabla_{\boldsymbol{\epsilon}}\left(\boldsymbol{\epsilon}^2 - \boldsymbol{\lambda}\boldsymbol{\epsilon}\right) \\ &= 2\boldsymbol{\epsilon} - \boldsymbol{\lambda} \\ \boldsymbol{\epsilon} &= \frac{1}{2}\boldsymbol{\lambda} \end{aligned} \tag{13}$$

### B.3 GRADIENT FOR $\psi$

First we derive the gradient of $\zeta_\psi$

$$\begin{aligned} \nabla_\psi \zeta_\psi &= \int \frac{p_\theta\left(y\right)}{p_\theta\left(y\right)} \nabla_\psi e^{f_\psi(y)} \mathrm{d}y \\ &= \int p_\theta\left(y\right) e^{f_\psi(y) - \log p_\theta(y)} \nabla_\psi f_\psi\left(y\right) \mathrm{d}y \\ &= \mathbb{E}_{y \sim p_\theta}\left[e^{f_\psi(y) - \log p_\theta(y)} \nabla_\psi f_\psi\left(y\right)\right] \\ &\approx \frac{1}{M}\sum_{i=1}^{M} \left[e^{f_\psi(y_i) - \log p_\theta(y_i)} \nabla_\psi f_\psi\left(y_i\right)\right]. \end{aligned} \tag{14}$$

We use this for deriving the gradient of the Lagrangian

$$
\begin{aligned}
\nabla_\psi \mathcal{L} &= \lambda_{\mathrm{prx}} \frac{1}{N} \sum_{i=1}^{N} \left[ -\nabla_\psi f_\psi (x_i) \right] + \lambda_{\mathrm{rv}} \frac{1}{N} \sum_{i=1}^{N} \left[ -\nabla_\psi f_\psi (y_i) \right] \\
&\quad + \left( \frac{\lambda_{\mathrm{rv}} + \lambda_{\mathrm{prx}}}{\zeta_\psi} + \lambda_{\zeta,u} - \lambda_{\zeta,l} \right) \nabla_\psi \zeta_\psi \\
&= \frac{1}{N} \sum_{i=1}^{N} \left[ -\lambda_{\mathrm{prx}} \nabla_\psi f_\psi (x_i) \right] + \frac{1}{N} \sum_{i=1}^{N} \left[ -\lambda_{\mathrm{rv}} \nabla_\psi f_\psi (y_i) \right] \\
&\quad + \left( \frac{\lambda_{\mathrm{rv}} + \lambda_{\mathrm{prx}}}{\zeta_\psi} + \lambda_{\zeta,u} - \lambda_{\zeta,l} \right) \frac{1}{N} \sum_{i=1}^{N} \left[ e^{f_\psi(y_i) - \log p_\theta(y_i)} \nabla_\psi f_\psi (y_i) \right] \\
&= \frac{1}{N} \sum_{i=1}^{N} \left[ -\lambda_{\mathrm{prx}} \nabla_\psi f_\psi (x_i) \right] \\
&\quad + \frac{1}{N} \sum_{i=1}^{N} \left[ \left( \left( \frac{\lambda_{\mathrm{rv}} + \lambda_{\mathrm{prx}}}{\zeta_\psi} + \lambda_{\zeta,u} - \lambda_{\zeta,l} \right) e^{f_\psi(y_i) - \log p_\theta(y_i)} - \lambda_{\mathrm{rv}} \right) \nabla_\psi f_\psi (y_i) \right]
\end{aligned}
\tag{15}
$$

where $\zeta_\psi$ is estimated as in Equation (12).

### B.4 GRADIENT FOR $\theta$

We first derive the helper gradients

$$
\begin{aligned}
\nabla_\theta \mathbb{E}_{y \sim p_\theta} \left[ f_\psi (y) \right] &= \nabla_\theta \int f_\psi (y) \, p_\theta (y) \, \mathrm{d}y \\
&= \int f_\psi (y) \, p_\theta (y) \, \nabla_\theta \log p_\theta (y) \, \mathrm{d}y \\
&= \mathbb{E}_{y \sim p_\theta} \left[ f_\psi (y) \, \nabla_\theta \log p_\theta (y) \right] \\
&\approx \frac{1}{N} \sum_{i=1}^{N} \left[ f_\psi (y_i) \, \nabla_\theta \log p_\theta (y_i) \right]
\end{aligned}
\tag{16}
$$

and

$$
\begin{aligned}
\nabla_\theta \mathbb{E}_{y \sim p_\theta} \left[ \log p_\theta (y) \right] &= \nabla_\theta \int p_\theta (y) \log p_\theta (y) \, \mathrm{d}y \\
&= \int \log p_\theta (y) \, \nabla_\theta p_\theta (y) + p_\theta (y) \, \nabla_\theta \log p_\theta (y) \, \mathrm{d}y \\
&= \int p_\theta (y) \cdot \log p_\theta (y) \, \nabla_\theta \log p_\theta (y) \, \mathrm{d}y + \int \nabla_\theta p_\theta (y) \, \mathrm{d}y \\
&= \mathbb{E}_{y \sim p_\theta} \left[ \log p_\theta (y) \, \nabla_\theta \log p_\theta (y) \right] + \nabla_\theta \int \cancel{p_\theta (y)} \, \mathrm{d}y \\
&= \mathbb{E}_{y \sim p_\theta} \left[ \log p_\theta (y) \, \nabla_\theta \log p_\theta (y) \right] \\
&\approx \frac{1}{N} \sum_{i=1}^{N} \left[ \log p_\theta (y_i) \, \nabla_\theta \log p_\theta (y_i) \right],
\end{aligned}
\tag{17}
$$

where the last term is removed because the integral over a probability distribution is unity, and the gradient of a constant is zero. We use these for gradient of the Lagrangian

$$
\begin{aligned}
\nabla_\theta \mathcal{L} &= \nabla_\theta \left( \frac{1}{N} \sum_{i=1}^N \left[ -\lambda_{\mathrm{fw}} \log p_\theta \left( x_i \right) \right] + \lambda_{\mathrm{rv}} \frac{1}{N} \sum_{i=1}^N \left[ \log p_\theta \left( y_i \right) - f_\psi \left( y_i \right) \right] \right) \\
&= \frac{1}{N} \sum_{i=1}^N \left[ -\lambda_{\mathrm{fw}} \nabla_\theta \log p_\theta \left( x_i \right) \right] + \lambda_{\mathrm{rv}} \nabla_\theta \frac{1}{N} \sum_{i=1}^N \left[ \log p_\theta \left( y_i \right) \right] - \lambda_{\mathrm{rv}} \frac{1}{N} \sum_{i=1}^N \left[ f_\psi \left( y_i \right) \right] \\
&= \frac{1}{N} \sum_{i=1}^N \left[ -\lambda_{\mathrm{fw}} \nabla_\theta \log p_\theta \left( x_i \right) \right] + \lambda_{\mathrm{rv}} \frac{1}{N} \sum_{i=1}^N \left[ \log p_\theta \left( y_i \right) \nabla_\theta \log p_\theta \left( y_i \right) \right] \\
&\quad - \lambda_{\mathrm{rv}} \frac{1}{N} \sum_{i=1}^N \left[ f_\psi \left( y_i \right) \nabla_\theta \log p_\theta \left( y_i \right) \right] \\
&= \frac{1}{N} \sum_{i=1}^N \left[ -\lambda_{\mathrm{fw}} \nabla_\theta \log p_\theta \left( x_i \right) \right] + \frac{1}{N} \sum_{i=1}^N \left[ \lambda_{\mathrm{rv}} \left( \log p_\theta \left( y_i \right) - f_\psi \left( y_i \right) \right) \nabla_\theta \log p_\theta \left( y_i \right) \right]
\end{aligned}
\tag{18}
$$

### B.5 LAGRANGE MULTIPLIERS GRADIENTS

The gradients of the Lagrange multipliers $\boldsymbol{\lambda}$ are relatively straightforward:

$$
\begin{aligned}
\nabla_{\lambda_{\mathrm{fw}}} \mathcal{L} &= \frac{1}{N} \sum_{i=1}^N \left[ -\log p_\theta \left( x_i \right) \right] - \epsilon_{\mathrm{fw}} \\
\nabla_{\lambda_{\mathrm{rv}}} \mathcal{L} &= \frac{1}{N} \sum_{i=1}^N \left[ \log p_\theta \left( y_i \right) - f_\psi \left( y_i \right) + \log \zeta_\psi \right] - \epsilon_{\mathrm{rv}} \\
\nabla_{\lambda_{\mathrm{prx}}} \mathcal{L} &= \frac{1}{N} \sum_{i=1}^N \left[ -f_\psi \left( y_i \right) + \log \zeta_\psi \right] - \epsilon_{\mathrm{prx}} \\
\nabla_{\lambda_{\mathrm{u}}} \mathcal{L} &= \frac{1}{M} \sum_{i=1}^M \left[ e^{f_\psi(y_i) - \log p_\theta(y_i)} \right] - 1 - \epsilon_\zeta \\
\nabla_{\lambda_\ell} \mathcal{L} &= 1 - \frac{1}{M} \sum_{i=1}^M \left[ e^{f_\psi(y_i) - \log p_\theta(y_i)} \right]
\end{aligned}
\tag{19}
$$

---

**Algorithm 2** Estimation of $\log \zeta_\psi$ according to Equation (5)

**Require:** An EBM $f_\psi$, a NF $p_\theta$, number of samples $m$
  Sample $\{x_i\}_1^m$ from $p_\theta$
  Compute the log probabilities $\{\log p_\theta \left( x_i \right)\}_1^m$
  Compute the EBM outputs $\{f_\psi \left( x_i \right)\}_1^m$
  $\log \zeta_\psi \leftarrow \mathrm{LogSumExp} \left( \{f_\psi \left( x_i \right) - \log p_\theta \left( x_i \right)\}_1^m \right) - \log m$
  Return $\log \zeta_\psi$

---

---

**Algorithm 3** Optimization algorithm for Equation (D̂-DYN)

---

**Require:** Datasets $\mathcal{D}_{\text{data}}$, learning rates $\alpha$, batch size $B$, number of epochs $E$
  **for** $e = 1$ to $E$ **do**
    **for** each minibatch $\mathcal{B}_{\text{data}} \subseteq \mathcal{D}_{\text{data}}$ **do**
      **StopGradient:** Sample $|\mathcal{B}_{\text{data}}|$ samples $x$ from $p_\theta$
      **StopGradient:** Estimate $\log \zeta_\psi$ from Algorithm 2

      Compute the log probabilities $\log p_\theta(x)$
      Compute the output for the data samples $f_\psi(\mathcal{B}_{\text{data}})$ and $p_\theta$ samples $f_\psi(x)$
      Compute the log probability of the data under $p_\theta$ $\log p_\theta(\mathcal{B}_{\text{data}})$

      Solve $u$ according to Equation (13)
      Compute $\nabla_\psi \mathcal{L}$ from Equation (15)
      Compute $\nabla_\theta \mathcal{L}$ from Equation (18)
      Compute $\nabla_\lambda \mathcal{L}$ from Equation (19)
      Run optimizer step and update $\psi, \theta$ with gradient descent, and $\boldsymbol{\lambda}$ with gradient ascent
      **StopGradient:** Clamp $\boldsymbol{\lambda}$ to be non-negative $\boldsymbol{\lambda} \leftarrow \max\{\mathbf{0}, \boldsymbol{\lambda}\}$
    **end for**
  **end for**
  **return** $\psi^*, \theta^*$

---

### B.6 CONDITIONAL FORMULATION

We use the following formulation for conditional probabilities

$$\operatorname*{minimize}_{\psi, \theta, \boldsymbol{\epsilon}} \quad \epsilon_{\text{fw}}^2 + \epsilon_{\text{rv}}^2 + \epsilon_{\text{EBM}}^2$$

$$\text{subject to} \quad -\frac{1}{N} \sum_{i=1}^{N} \log p_\theta(\beta_i | u_i) \leq \epsilon_{\text{fw}}$$

$$\frac{1}{N} \sum_{i=1}^{N} D_{\text{KL}}\left(p_\theta(\cdot | u_i) \,\|\, q_\psi(\cdot | u_i)\right) \leq \epsilon_{\text{rv}} \qquad \text{(PIII)}$$

$$-\frac{1}{N} \sum_{i=1}^{N} \left[\log f_\psi(\beta_i | u_i) - \log \zeta_\psi^{(\beta_i)}\right] \leq \epsilon_{\text{EBM}}$$

$$1 \leq \frac{1}{N} \sum_{i=1}^{N} \zeta_\psi^{(\beta_i)} \leq 1 + \epsilon_\zeta,$$

where $\gamma_i$ are generated by $p_\theta$. Since each condition has a different partition function, we choose to take the empirical mean of partitions over the given conditions.

### B.7 NEGATIVE NLL

Note that for the simplified problem

$$\operatorname*{minimize}_{\psi, \epsilon} \quad \epsilon^2$$
$$\text{subject to} \quad f(\psi) \leq \epsilon, \qquad \text{(PIV)}$$

the NLL can never have a negative upper bound due to the update rule of $u$ (Equation (13)) and the fact that $\boldsymbol{\lambda}$ are always non-negative. But in certain cases, a model can attain a negative NLL. In these cases we use a more generalized form of resilient learning as

$$\operatorname*{minimize}_{\psi, \epsilon, \delta} \quad \epsilon^2 - \delta$$
$$\text{subject to} \quad f(\psi) \leq \epsilon - \delta^2. \qquad \text{(PV)}$$

This revised problem now allows the upper bound of the NLL to be negative, as $v$ is maximized which can make the upper bound in the constraint negative. In this case, all gradients are unchanged and $u$ retains its closed form solution (Equation (13)). Solving $\delta$ also has a closed form solution

$$
\begin{aligned}
0 &= \nabla_\delta \mathcal{L} \\
&= \nabla_\delta \left[ \epsilon^2 - \delta + \lambda \left( f\left(\psi\right) - \epsilon + \delta^2 \right) \right] \\
&= -1 + 2\lambda\delta \\
\delta &= \frac{1}{2\lambda}.
\end{aligned}
\tag{20}
$$

Note that for the NLL and KL, the constraint is always active, preventing $\lambda$ from being zero and $v$ from being undefined. When using this form, we also initialize $\boldsymbol{\lambda}$ to be non-zero. The full revised problem is then

$$
\begin{aligned}
\min_{\psi,\theta,\boldsymbol{\epsilon},\boldsymbol{\delta}} \quad & \epsilon_{\text{fw}}^2 - \delta_{\text{fw}} + \epsilon_{\text{rv}}^2 + \epsilon_{\text{EBM}}^2 - \delta_{\text{EBM}} \\
\text{subject to} \quad & \text{NLL}\left(p_\theta\right) \leq \epsilon_{\text{fw}} - \delta_{\text{fw}}^2 \\
& D_{\text{KL}}\left(p_\theta \parallel q_\psi\right) \leq \epsilon_{\text{rv}} \\
& \text{NLL}\left(q_\psi\right) \leq \epsilon_{\text{EBM}} - \delta_{\text{EBM}}^2 \\
& 1 \leq \zeta_\psi \leq 1 + \epsilon_\zeta.
\end{aligned}
\tag{PVI}
$$

### B.8 TRAINING IN HIGH DIMENSIONS

In high dimensions, the initial values of $p_\theta$ and $f_\psi$ can be in different orders of magnitude. The initial outputs of the MLP $f_\psi$ are usually small around zero while the log probabilities of $p_\theta$ are larger negative values the higher the dimension. These differences can lead to exploding gradients due to the exponent term in Equation (15). To mitigate this difference in order of magnitude, we add two mechanisms. First, we add a temperature $T$ to the EBM, which makes it easier for the EBM to output high negative values

$$
q_\psi\left(x\right) = \frac{1}{\zeta_\psi} e^{\frac{1}{T} f_\psi(x)}.
\tag{21}
$$

Then, before starting training, we warm start $f_\psi$ to produce values similar to the log probabilities of $p_\theta$ in an iterative process. In each iteration, we sample from $p_\theta$ with the log probabilities, compute the $f_\psi$ values on the samples and minimize the mean squared error between the the two sets of corresponding values. With these two additions, training is starting in a stable manner with less risk of exploding gradients.

We also replace the KL term between $p_\theta$ and $q_\psi$ with a weighted KL using the same temperature $T$ as

$$
\begin{aligned}
T \cdot D_{\text{KL}}\left(p_\theta \parallel q_\psi\right) &= T\mathbb{E}_{x \sim p_\theta}\left[ \log p_\theta\left(x\right) - \frac{1}{T} f_\psi\left(x\right) + \log \zeta_\psi \right] \\
&= \mathbb{E}_{x \sim p_\theta}\left[ T \log p_\theta\left(x\right) - f_\psi\left(x\right) + T \log \zeta_\psi \right],
\end{aligned}
\tag{22}
$$

which avoids the large values of $p_\theta$.

## C IMPLEMENTATION DETAILS

### C.1 MODEL ARCHITECTURES

For NF and WGAN's generators, we use a neural spline flow (Durkan et al., 2019), implemented with the Zuko library[2]. For the EBMs and WGAN's critic we used a residual MLP comprising several residual blocks as described in Algorithm 4. Each residual block consists of several simple layers, as detailed in Algorithm 5. For the conditional case, we use the same architecture as the residual MLP, but replace the block with residual FiLM blocks (Perez et al., 2018), as detailed in Algorithm 6.

---

[2]https://zuko.readthedocs.io/stable/

---

**Algorithm 4** Residual MLP

---

**Require:** Input dimension $d$, hidden dimension $h$, number of blocks $n$
**Require:** Input vector $x \in \mathbb{R}^d$
   $x' \leftarrow \text{Linear}\,(x)$         $\triangleright$ Linear layer with output dimension $h$ with spectral normalization
   $x' \leftarrow \text{SiLU}\,(x')$              $\triangleright$ Sigmoid Linear Unit activation
   **for** $i = 1$ to $n$ **do**
      $x' \leftarrow \text{ResidualBlock}\,(x')$       $\triangleright$ Residual block as in Algorithm 5 with dimension $h$
   **end for**
   $x' \leftarrow \text{Linear}\,(x')$         $\triangleright$ Linear layer with input dimension $h$ and output dimension 1
   Return $x'$

---

**Algorithm 5** Residual block

---

**Require:** Dimension $d$
**Require:** Input vector $x \in \mathbb{R}^d$
   $x' \leftarrow \text{Layer Normalization}\,(x)$
   $x' \leftarrow \text{SiLU}\,(x')$              $\triangleright$ Sigmoid Linear Unit activation
   $x' \leftarrow \text{Linear}\,(x)$        $\triangleright$ Linear layer with input and output dimension $d$ and spectral norm
   $x' \leftarrow \text{Layer Normalization}\,(x)$
   $x' \leftarrow \text{SiLU}\,(x')$              $\triangleright$ Sigmoid Linear Unit activation
   $x' \leftarrow \text{Linear}\,(x)$        $\triangleright$ Linear layer with input and output dimension $d$ and spectral norm
   $x' \leftarrow \alpha x + x'$           $\triangleright$ Residual connection with trainable parameter $\alpha$
   $x' \leftarrow \text{SiLU}\,(x')$
   Return $x'$

---

**Algorithm 6** Conditional residual block

---

**Require:** Dimension $d$, number of FiLM layers $m$
**Require:** Input vector $x \in \mathbb{R}^d$, condition vector $c \in \mathbb{R}^k$
   $\gamma \leftarrow \text{MLP}\,(c)$
       $\triangleright$ MLP with input dimension $k$ and output dimension $d$, with $m$ layers, SiLU activation

   $\beta \leftarrow \text{MLP}\,(c)$
       $\triangleright$ MLP with input dimension $k$ and output dimension $d$, with $m$ layers, SiLU activation

   $x' \leftarrow \text{MLP}\,(x)$
       $\triangleright$ MLP with input dimension $d$ and output dimension $d$, with 2 layers, SiLU activation

   $x' \leftarrow \gamma \odot x' + \beta$           $\triangleright$ $\odot$ represents element-wise multiplication
   $x' \leftarrow \text{SiLU}\,(\alpha x + x')$       $\triangleright$ Residual connection with trainable parameter $\alpha$
   Return $x'$

---

## C.2 AUTOENCODER

To reduce the dimensionality of CelebA, we use a autoencoder with a 100D latent space. We use the 64X64 version of the dataset, and scale the pixel values to be between -1 and 1. We used Algorithm 7 for the encoder and Algorithm 8 for the decoder, and train for 20 epochs with a batch size of 128 and a learning rate of $10^{-3}$ using Adam. The objective for the autoencoder was the mean square error (MSE) loss between the input and reconstructed output

---

**Algorithm 7** Encoder

---

**Require:** Latent dimension $d$
**Require:** Input vector $x \in \mathbb{R}^D$
  $x' \leftarrow \text{Conv2D}(x)$     ▷ Convolution layer with 32 channels, kernel size 4, stride 2 and padding 1
  $x' \leftarrow \text{ReLU}(x')$
  $x' \leftarrow \text{Conv2D}(x')$     ▷ Convolution layer with 64 channels, kernel size 4, stride 2 and padding 1
  $x' \leftarrow \text{ReLU}(x')$
  $x' \leftarrow \text{Conv2D}(x')$     ▷ Convolution layer with 128 channels, kernel size 4, stride 2 and padding 1
  $x' \leftarrow \text{ReLU}(x')$
  $x' \leftarrow \text{Conv2D}(x')$     ▷ Convolution layer with 256 channels, kernel size 4, stride 2 and padding 1
  $x' \leftarrow \text{ReLU}(x')$
  $x' \leftarrow \text{MLP}(x')$                           ▷ MLP with output dimension $d$
  Return $x'$

---

**Algorithm 8** Decoder

---

**Require:** Input vector $x \in \mathbb{R}^d$
  $x' \leftarrow \text{MLP}(x)$                      ▷ MLP with output dimension $256 \times 4 \times 4$
  $x' \leftarrow \text{ConvT2D}(x')$   ▷ Transposed convolution: 64 channels, kernel size 4, stride 2, padding 1
  $x' \leftarrow \text{ReLU}(x')$
  $x' \leftarrow \text{ConvT2D}(x')$   ▷ Transposed convolution: 32 channels, kernel size 4, stride 2, padding 1
  $x' \leftarrow \text{ReLU}(x')$
  $x' \leftarrow \text{ConvT2D}(x')$     ▷ Transposed convolution: 3 channels, kernel size 4, stride 2, padding 1
  Return $x'$

---

## C.3 OPTIMIZATION AND EVALUATION

All our code is written in Python using PyTorch[3] and Lightning[4]. For optimization we use Adam with parameters $(\beta_1, \beta_2) = (0.0, 0.9)$ and change the learning rate and batch size per experiment.

# D EXPERIMENTAL DETAILS AND RESULTS

## D.1 COMPARISON ON 40-COMPONENT GAUSSIAN MIXTURE

In Section 4 we compare our method with the weighted sum formulation, NF and WGAN. The 2D distribution we used for the comparison, inspired by Midgley et al. (2023), consisted of a Gaussian mixture with 40 components. The center of each component is drawn uniformly between 0 and 1, and each component has a standard deviation of 0.05. For the train set we used 800 samples, and for the test set 10,000 samples. We trained all models for 100,000 iterations with full batches of the whole train set. For the Lagrange multipliers we used a learning rate of $10^{-4}$, while the lagrange multipliers corresponding to the $\zeta_\psi$ normalization constraints were trained with a learning rate of $10^{-3}$ where $\epsilon_\zeta = 0.1$. Since some of the shapes have low entropies, we used the formulation in Appendix B.7 in all datasets.

For NF and our $p_\theta$ we used a neural spline flow (Appendix C.1) with 5 transforms and 8 bins. For our $q_\psi$ and WGAN's critic we used a residual MLP (Algorithm 4) with 6 residual blocks and a hidden dimension of 64.

---

[3] https://pytorch.org/
[4] https://lightning.ai/docs/pytorch/stable/

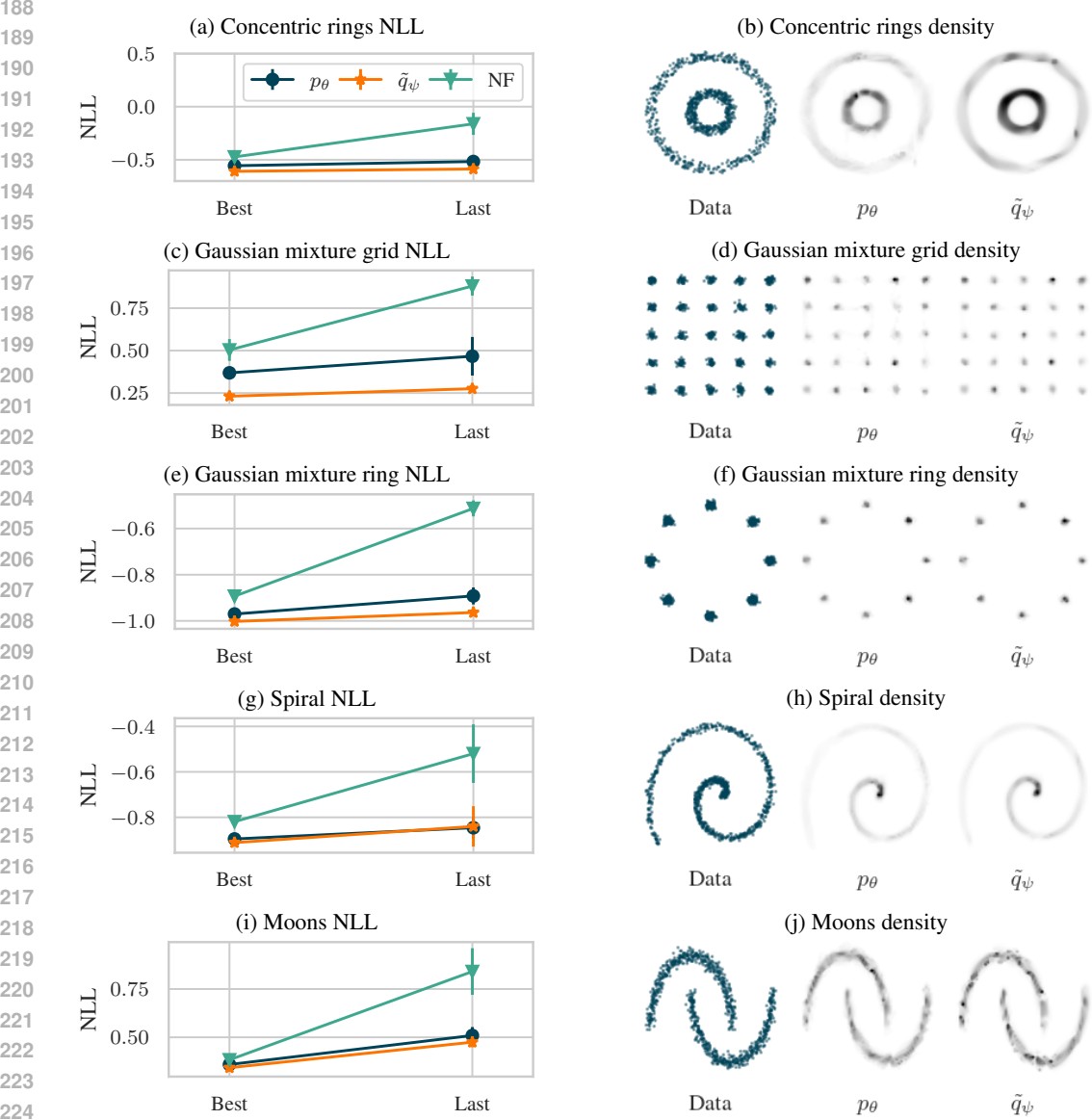

Figure 7: Our framework is able to accurately learn the density of various 2D datasets. The left column presents the lowest NLL each model achieved during training and the NLL at the end of training, demonstrating that our method outperforms NF in terms of values and consistency. The right column qualitatively displays the finite dataset (left) and the learned density values from of $p_\theta$ and the quasi-normalized $\tilde{p}_\psi$ (middle and right respectively). The qualitative density maps show that both models perfectly capture the shape and all modes.

In the comparison with the weighted sum formulation (Equation (P-W)) we used 25 different weight configurations, where each weight $w_{\text{back}}, w_{\text{for}} \in \{0.2, 0.5, 1.0, 2.0, 5.0\}$. For both formulations, we used a learning rate (LR) of $10^{-3}$ for $p_\theta$ and $10^{-4}$ for $q_\psi$, and we estimated $\zeta_\psi$ with $M = 1000$ samples from $p_\theta$. We trained the NF comparison also with a LR of $10^{-3}$.

When comparing with WGAN, we used a reduced LR for $q_\psi$ and the critic of $10^{-5}$ to stabilize training, and trained $p_\theta$ and the generator with a LR of either $10^{-3}$ or $10^{-5}$. We trained the WGAN with gradient penalty (Gulrajani et al., 2017) with a coefficient of 0.1.

## D.2 Density estimation

In the density estimation experiments in Section 5.2 we tested 5 different distributions, which can be seen in Figure 7. For each dataset we used a train set of 1000 samples and a test set of 10000 samples and trained for 25000 full-batch iterations. We used a LR of $10^{-3}$ for $p_\theta$ and NF, and LR of $10^{-4}$ for $q_\psi$. We estimated $\zeta_\psi$ with $M = 1000$ samples from $p_\theta$. For the Lagrange multipliers we used a learning rate of $10^{-4}$, while the lagrange multipliers corresponding to the $\zeta_\psi$ normalization constraints were trained with a learning rate of $10^{-3}$, where $\epsilon_\zeta = 0.1$. The initial values for the KL Lagrange multipliers was set to 0.01

To create the concentric rings dataset, we sample $N$ points from a standard normal distribution, normalize them to have unit norm, and scale them by a either $r_1 = 0.2$ or $r_2 = 0.6$ with 0.5 probability. We then added uniform random noise $\mathcal{U}(0, 0.1)$ to each point. To create the Gaussian mixture grid, we created a Gaussian mixture distribution of a 5x5 grid of centers between -1 and 1 which we then scaled to have zero mean and unit variance, each with a standard deviation of 0.05. To create the Gaussian mixture ring, we created a Gaussian mixture distribution of a 8 components equidistant on a unit circle which we then scaled to have zero mean and unit variance. Each component has a standard deviation of 0.05. To create the spiral distribution, we sampled $N$ points uniformly $t \sim \mathcal{U}(0, 10)$, and created points $(x, y)$ with $x = t \cos(t)$ and $y = t \sin(t)$. We then added Gaussian noise with standard deviation of 0.02 to each point. For the moons dataset we used the "make moons" function from the scikit-learn package[5].

To draw the qualitative results, we divided the visualized region into $64 \times 64$ pixels and computed the density value of each pixel as $v = p(x) \cdot A$, with $A$ being the area of each pixel. We then plot the filled contours of the density values.

## D.3 Image sampling

We train our models on the encoded 100D space of CelebA (Liu et al., 2015). We use an autoencoder to reduce dimensionality as described in Appendix C.2. For $p_\theta$ and NF we used a neural spline flow (Appendix C.1) with 6 transforms and 16 bins. For $q_\psi$ we used a residual MLP (Algorithm 4) with 5 residual blocks and a hidden dimension of 1024. For training our models, we used a temperature 0.01, as described in Appendix B.8, and used 5000 samples to estimate $\zeta_\psi$. To save time, we sampled the 5000 samples only every 100 iterations. We trained all models for 40 epochs with batch size 128. The LR for $p_\theta$ and NF was $10^{-3}$, and for $q_\psi$ it was $10^{-6}$. After every epoch, we evaluated the FID[6] with 50,000 samples and report the best FID that $p_\theta$ and NF achieved during training. For the Lagrange multipliers we used a learning rate of $10^{-6}$, while the lagrange multipliers corresponding to the $\zeta_\psi$ normalization constraints were trained with a learning rate of $10^{-4}$, where $\epsilon_\zeta = 100$.

## D.4 Simulation-based inference

For SBI, we used two benchmark datasets: two moons (Greenberg et al., 2019), Gaussian mixture (Sisson et al., 2007). Each benchmark defines a prior and a simulator.

**Gaussian mixture** The parameter dimension is $\beta \in \mathbb{R}^2$ and the outcome dimension is $u \in \mathbb{R}^2$. The prior is a uniform distribution $p_\beta = \mathcal{U}\left([0.5, 1]^2\right)$ and the simulator is a Gaussian mixture of 2 components $p(x|\beta) = 0.5\mathcal{N}(\beta, I) + 0.5\mathcal{N}(\beta, 0.01I)$, where $I$ is the identity matrix.

**Two moons** The parameter dimension is $\beta \in \mathbb{R}^2$ and the outcome dimension is $u \in \mathbb{R}^2$. The prior is a uniform distribution $p_\beta = \mathcal{U}\left([-1, 1]^2\right)$. The simulator is

$$u|\beta = (r\cos(\alpha) + 0.25, r\sin(\alpha)) + \left(-|\beta_1 + \beta_2|/\sqrt{2}, (-\beta_1 + \beta_2)/\sqrt{2}\right),$$

where $\alpha \sim \mathcal{U}\left(-\frac{\pi}{2}, \frac{\pi}{2}\right)$ and $r \sim \mathcal{N}\left(0.1, 0.01^2\right)$.

---

[5]https://scikit-learn.org/
[6]We adapted the code from https://github.com/mseitzer/pytorch-fid.

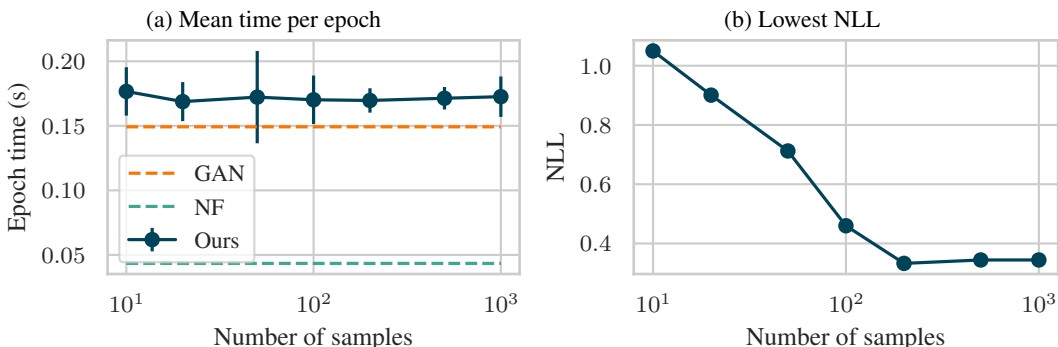

Figure 8: How the number of samples for the estimation of $\zeta_\psi$ with importance sampling (Equation (5)) affect training. (a) The computation time for our method is slower than NF and GANs, but does not scale much with more samples. (b) Using more samples improves the NLL.

To evaluate the C2ST, we generate 5000 samples from the posterior using rejection sampling. For each dataset we defined a test condition $u_0$ and generated samples by sampling from the prior, running the simulator and keep the first 5000 samples that were close to $u_0$. For the Gaussian mixture we set $u_0 = (0.75, 0.75)$ and for the two moons $u_0 = (0, 0)$. We also generate 5000 samples from the trained models by conditioning on $u_0$. We split the total of 10000 to a train and test set with a 0.3 split and train a classifier[7] with positive labels for the ground truth points and negative labels for the model points, and use its accuracy on the test set as the C2ST metric.

To generate from $q_\psi$, we followed the MCMC sampling in Equation (8). We used 2000 steps with a step size of $10^{-4}$. The initial points were sampled from $p_\theta$.

For $p_\theta$ and NF we used with 4 transforms and 8 bins. For $q_\psi$ we used a different architecture for each dataset. For the Gaussian mixture, we used a conditional residual MLP (Algorithm 6) with 6 residual blocks and a hidden dimension of 64 and 2 FiLM layers. For the two moons, we used 3 residual blocks, each with 128 channels and 2 FiLM layers. We trained with different number of simulations (data points), but used a batch size of 1000 for all experiments. We used the negative NLL formulation (Appendix B.7) for both datasets, with LR of $10^{-4}$ for $p_\theta$ and NF, and LR of $10^{-5}$ for $q_\psi$. The Lagrange multipliers LR was set to $10^{-6}$, while the lagrange multipliers corresponding to the $\zeta_\psi$ normalization constraints were trained with a learning rate of $10^{-2}$, where $\epsilon_\zeta = 30$. The initial values for the KL Lagrange multipliers was set to 0.01.

# E  COMPUTE TIME

Our method requires generating $M$ samples to estimate the partition function $\zeta_\psi$ with importance sampling (Equation (5)). Here we study how the number of samples $M$ affects the training time and performance. We used our training data from the 40 components Gaussian mixture experiment from Section 5.1. We trained all models on an NVIDIA A100 GPU and averaged the time per epoch over 100,000 epochs.

Figure 8a shows that our method is indeed slower than NF, but not much slower than GANs. It is also interesting to observe that the compute time barely increases with more samples. This is because the importance sampling does not require gradients, and with the stop gradient (Algorithm 3), the computation is efficient. For a much larger number of samples, or higher dimensions, we expect the time per epoch to increase.

We also examine how the number of samples affects the quality of learning. In Figure 8b we plot the best NLL achieved during training per number of samples. It is clear that with too few samples, the inaccuracy in the estimation harms learning.

---

[7]HistGradientBoostingClassifier from https://scikit-learn.org

