# OpenReview forum: "Adaptive Symmetrization of the KL Divergence"
_ICLR.cc/2026/Conference — Submitted to ICLR 2026_

### Official Review · Reviewer_Nftq · 2025-10-25

**Soundness:** 1
**Presentation:** 3
**Contribution:** 2
**Rating:** 2
**Confidence:** 4

**Summary:**

This paper proposes a method to minimize the symmetric Jeffreys divergence (the sum of forward and reverse KL divergences) to learn a probability distribution from data. The key challenge is that the reverse KL term depends on the unknown data distribution. To address this, the authors introduce a proxy model to approximate the data distribution and formulate the problem as a constrained optimization task. A central contribution is an "adaptive symmetrization" mechanism, implemented via a resilient optimization framework (P-DYN), which dynamically adjusts the emphasis on the forward KL, reverse KL, and proxy model fidelity during training. The authors develop a primal-dual algorithm based on non-convex duality theory to solve this problem and propose a synergistic combination of a Normalizing Flow (NF) as the primary model and an Energy-Based Model (EBM) as the proxy. Experimental results on synthetic 2D data, latent-space image sampling (CelebA), and simulation-based inference (SBI) benchmarks are provided, claiming improved stability and performance over baselines like NF, WGAN, and a fixed-weight penalty method.

**Strengths:**

The core idea of using a collaboratively trained proxy model to enable the minimization of the Jeffreys divergence is novel and interesting. Moving away from the adversarial setup of GANs towards a collaborative, constrained optimization framework is a worthwhile direction.

**Weaknesses:**

The paper lacks a critical discussion on the fundamental tension between the forward and reverse KL divergences. It is well-established that their minima can be contradictory (mode-covering vs. mode-seeking). The proposed adaptive symmetrization aims to balance them, but the theoretical conditions under which minimizing their sum leads to a desirable solution are not analyzed. The claim that the method avoids the issues of both extremes needs deeper justification.

The transition from the idealized problem (PI) to the dynamically constrained problem (P-DYN) is presented as a solution for infeasibility, but the approximation gap between these two formulations is not quantified or analyzed. It remains unclear how the solutions of (P-DYN) relate to the original goal of minimizing the Jeffreys divergence.

The transition from Eq. (P-DYN) to the empirical dual problem in Eq. ($\hat P$-DYN) is flawed. The entropy of the data distribution is ignored. This oversight invalidates the equivalence claimed in this step.

The experiments are primarily conducted on low-dimensional, synthetic 2D datasets. While useful for illustration, they are insufficient to demonstrate the scalability and practical utility of the method for modern machine learning problems. The claim that the method is "more accurate on a variety of datasets" is overstated.

The comparisons, while including NF and WGAN, lack benchmarks against other state-of-the-art generative models (e.g., diffusion models, VAEs) or other methods for symmetric divergence minimization.

There are related work on combining forward and reverse KL, such as the $\alpha$-bridge [1]. Discussions on the pros and cons of the proposed method over these existing ones are necessary.


[1] Zhao, Miaoyun, et al. "Bridging maximum likelihood and adversarial learning via α-divergence." Proceedings of the AAAI Conference on Artificial Intelligence. Vol. 34. No. 04. 2020.

**Questions:**

Please refer to the Weaknesses.

---

> ### Author Response · Authors · 2025-11-20
>
> We thank the reviewer for their time and feedback.
>
> 1.  **Discussion on forward vs. reverse KL**: We respectfully disagree with the reviewer that the manuscript “lacks critical discussion on the fundamental tension between forward and reverse KL.” Indeed, we motivate our method using the fact that *“minimizing the forward KL divergence $D_{KL}(\pi ∥ p_{\theta})$ as in (P-KL), leads to solutions that cover the support of $\pi$  at the cost of overlooking sharp modes. In contrast, the reverse KL divergence $D_{KL}(p_{\theta} ∥ \pi)$  promotes a mode-seeking behavior that may ignore regions of $\pi$  with significant mass (Murphy, 2012)”* (line 131-134).
>
> 	   We further explore this tension when discussing the adaptivity of our method that favors one behavior or the other depending on the training stage. Explicitly, *“there could be phases during which the reverse KL approximation is inaccurate. In such situations, we may want to put more emphasis on the forward divergence in the objective of (PI) [... and] (temporarily) break the symmetry of the Jeffreys divergence [...]. Dynamically changing the focus between forward and reverse KL divergences steers training between mode-covering and mode-seeking behaviors”* (line 181-187).
>
> 	That being said, if the reviewer believes there are other specific aspects of these divergences that are important to be discussed to motivate our method, we can certainly include a remark on them in the paper.
>
>
> 2.  **Theoretical advantages of the Jeffreys divergence**: The “mode-seeking” and “mode-covering” behaviors are induced and not enforced by the different KL divergences. For instance, fitting one Gaussian to a mixture of two Gaussian sufficiently close together will yield the same result regardless of the direction (e.g., a mixture of two equally probable Gaussians with variance 1 and means 0 and 4, nicely visualized in https://www.tuananhle.co.uk/notes/reverse-forward-kl.html). To be precise, for the forward KL $D_{KL}(\pi ∥ p)$ to be finite, it must be that $\pi > 0 \Rightarrow p > 0$. For the reverse KL $D_{KL}(p ∥ \pi)$to be finite, the opposite holds, i.e., $\pi = 0 \Rightarrow p = 0$. Hence, the former enforces that the whole support of $\pi$ is covered, whereas the latter enforces that only the support is covered.
>
>     Minimizing the Jeffreys divergence $J(p,\pi)$ enforces both two conditions, i.e., it covers completely and exclusively the support of $\pi$. In this concrete sense, it precludes the issues that could arise from imposing only one of these conditions, namely, either mode-covering or mode-seeking behaviors.
>
>     If the reviewer believes these technical remarks would be beneficial in the manuscript, we would be happy to include them. It could also be that we misunderstood the reviewer’s point, in which case we would appreciate it if the reviewer were to explain what they mean concretely by "desirable solution.”

---

> > ### Author Response · Authors · 2025-11-20
> >
> > 3.  **Transition from (PI) to (P-DYN)**: (P-DYN) does not minimize the Jeffreys divergence nor does it attempt to. To be clear, (PI) is an idealized problem that cannot be effectively solved in the setting of the paper, namely, with only access to samples from $\pi$. Replacing $\pi$ by its empirical measure $\hat{\pi} = \frac{1}{N} \sum_n \delta(x-x_n)$ (where $\delta$ is the Dirac delta) has several downsides, chief among them is the fact that any solution of (PI) must cover completely and exclusively the support of $\hat{\pi}$ and, therefore, be discrete.
> >
> >     For that reason, we are forced to move away from (PI). As the manuscript argues, GANs “avoid this challenge by representing a symmetric divergence, such as the Jensen-Shannon divergence, as a variational problem [...]. This formulation results in a brittle min-max objective susceptible to diverging and an extreme mode-seeking behavior known as mode collapse” (line 142-145). We instead choose to keep with the Jeffreys divergence and approximate (PI) by (P-JD) (note that the Jeffreys divergence is an upper bound on the Jensen-Shannon divergence (Crooks & Sivak, 2011)). It is worth noting that for $\epsilon = 0$ (and without $h$), (PI) and (P-JF) are equivalent (they have the same optimal value). In practice, “depending on the expressiveness of the proxy model $q_{\psi}$ , it may be impossible to choose a small value for $\epsilon$” (lines 179-180). The approximation gap between (PI) and (P-JF) depends directly on the expressivity of $q_{\psi}$ and cannot be assessed in general.
> >
> >     The challenges of this expressivity limitation go beyond approximation. As we state in the manuscript, “unless we ensure that $q_{\psi}$ is close enough to $\pi$ throughout training, there could be phases during which the reverse KL approximation is inaccurate” (lines 181-182). (P-DYN) therefore modifies the objective of (PI) substantially as it no longer tackles the sum of forward and reverse KL (i.e., the Jeffreys divergence), but may “put more emphasis on the forward divergence in the objective of (P-JF), since it is computed directly from data. In other words, we may want to (temporarily) break the symmetry of the Jeffreys divergence and prioritize training $q_{\psi}$ to fit $\pi$” (line 183). This distinction between (P-DYN) and (PI) is crucial in our method and the fact that this symmetry breaking is done dynamically and automatically is key. Indeed, “it would be impractical to manually adjust these trade-offs during training, especially since they depend on the model classes $\mathcal{H}$, $\mathcal{H}_{\psi}$ and the data” (line 188). This is in contrast to (PIII) that uses fixed parameters, which in addition “is not equivalent to solving (P-DYN)” (line 214).
> >
> >
> >
> > 4.  **Transition from (P-DYN) to ($\hat{P}$-DYN)**: Before the revision the manuscript stated, we use “the relation between the forward KL divergence and the NLL from (P-NLL) to obtain an empirical version of (P-DYN)” (line 218). Nowhere do we claim that (P-DYN) is equivalent to ($\hat{P}$-DYN): it could not be, given that the latter is based on samples (empirical distribution). Still, note that the entropy of the data $H(\pi)$ can be incorporated into the values of $\epsilon_{fw}$ and $\epsilon_{prx}$ to recover (P-DYN). The reviewer has a point that this point was missing in the manuscript. We have included a remark to this effect in the revision (line 218-219).
> >
> >
> >
> > Crooks, Gavin E., and David A. Sivak. "Measures of trajectory ensemble disparity in nonequilibrium statistical dynamics." _Journal of Statistical Mechanics: Theory and Experiment_ 2011.06 (2011): P06003.

---

> > > ### Author Response · Authors · 2025-11-20
> > >
> > > 5.  **Experiments**: The reviewer has a point that our experiments focus on illustrating our methods in lower-dimensional settings. This limitation is explicitly discussed in section 5.3 and 6 and is one that is related to EBMs and NFs, not our underlying training method. Developing this novel training method is the main goal of this paper as opposed to development work needed to make it computationally scalable. That being said, we show image sampling results that, though admittedly preliminary, are certainly promising.
> > >
> > >     As we mention in the conclusion, “it would be interesting to extend this method to work in such scenarios [high dimensional settings] by estimating the probability of generated samples (Ben-Dov et al., 2024) or by using rectangular flows (Caterini et al., 2021)” (lines 476-478). We also suggest addressing this challenge by “using different models and divergences in (P-DYN). Strong candidates are score-based models with the Fisher divergence. There is a form of the Fisher divergence that avoids the score of $\pi$ by computing the Hessian” (line 483). Nevertheless, though we do consider different datasets (e.g., CelebA, SBI benchmarks), we agree that the correct claim is that the model is "more accurate on a variety of problems." We have corrected this expression in the revision.
> > >
> > >
> > > 6.  **Comparisons**: While our method can be considered a generative model, it has substantially different goals. VAEs maximize a lower bound of the log-likelihood and yields only samples, not an estimate of the underlying distribution (i.e., probability evaluation). The same is true for other generative models, e.g., diffusion models. In contrast, our algorithm directly optimizes the log-likelihood (more precisely, the Jeffreys divergence) and can provide not only samples (e.g., from the EBM proxy $q_\psi$), but also evaluate the probability of these samples (e.g., using the NF $p_\theta$).
> > >
> > >     Our experiments reflect this goal by evaluating not the “quality of samples,” but how well the underlying distribution is captured (focusing on methods that directly minimize the NLL). As such, we tackle a different problem than VAEs and believe that a comparison would be neither fair nor viable. In view of these points, we wonder if the reviewer believes a comparison to VAEs remains necessary and if so, how they envision such a comparison to work (e.g., how to compute probabilities from VAE).
> > >
> > >
> > >
> > > 7. **Relation to $\alpha$-bridge**: The reviewer has a point that there are close connections between the $\alpha$-bridge from Zhao et al. and our method. In particular, $\alpha$-bridge is a “geometric mean” of the forward and reverse KL divergences rather than the “arithmetic mean” used in the Jeffreys divergence (and consequently, our method). What is more, the transition between forward and reverse KL is done using a fixed $\alpha$ schedule in Zhao et al., whereas our method uses an adaptive method (often leading to non-monotonic behaviors). The goal of our method is not to transition from forward to reverse, but to balance the two. Finally, as opposed to Zhao et al., we do not rely on ELBO or adversarial methods to compute either directions of the KL divergence, using a proxy technique instead.
> > >
> > > 	We thank the reviewer for bringing this work to our attention. We have included these remarks in the related work of the manuscript in Section A.4.

---

### Official Review · Reviewer_sXKM · 2025-10-28

**Soundness:** 2
**Presentation:** 2
**Contribution:** 1
**Rating:** 2
**Confidence:** 5

**Summary:**

This paper proposes a new way to minimize the Jeffreys divergence by introducing a " proxy mode". The final objective is weighted combination of three KL divgerneces. The proposed method is then applied to task like: density estimation, image generation, and
simulation-based inference.

**Strengths:**

The experiment section covers different potential use cases of the proposal method, which is good.

**Weaknesses:**

1. **Motivation is unclear**
The motivation for using Jeffreys divergence is "training on discrete samples may lead to a
mismatch between the modelled distribution and the data distribution (illustrated in Figure 1a). Ordinarily, minimizing a symmetric divergence would alleviate this issue," which is unclear to me. I am not sure why minimising a symmetric divergence will make the mode and target distribution more matched. They are all valid divergences; any valid divergence will make two distributions equal when the divergence goes to zero. Need more explanation on the motivation, practical evidence or reference to illustrate this problem.

There are some benefits of combining forward KL and reverse KL for training, for example, in this paper https://arxiv.org/pdf/1907.11891, the motivation is that Reverse KL will lead to mode collapse but will get sharper mode estimation, forward KL will have better mode covering ability, so adding FKL to RKL will improve the diversity. This is one example of a valid motivation. This is the most important question to answer when starting research.

2. **The experiment results are too bad** The FID for CelebA is too high, and only on low low-dimensional latent space is too out of date. A valid paper either has a new method with a good motivation or can improve some current best methods. This FID is too high, couldn't show the proposed method is effective in high dimensions. Other 2d experiments are too trivial.

3. **Idea is not inspiring** The proposed method requires introducing another proxy model, which needs to be as powerful as the main model, which is unaffordable in current machine learning world.

**Questions:**

See above for the weekness.

---

> ### Author Response · Authors · 2025-11-20
>
> We thank the reviewer for their time in reviewing the paper. We address their concerns point-by-point below.
>
> 1.  **Clarity of motivation**: We fully agree with the motivation suggested by the reviewer and it is in fact the one used in the manuscript (line 131-134), namely, that *“minimizing the forward KL divergence $D_{KL}(\pi ∥ p_{\theta})$ as in (P-KL), leads to solutions that cover the support of $\pi$ at the cost of overlooking sharp modes. In contrast, the reverse KL divergence $D_{KL}(p_{\theta} ∥ \pi)$ promotes a mode-seeking behavior that may ignore regions of $\pi$ with significant mass (Murphy, 2012).”*
>
>     We further explore this dichotomy when discussing the adaptivity of our method that favors one behavior or the other depending on the training stage. Explicitly, *“there could be phases during which the reverse KL approximation is inaccurate. In such situations, we may want to put more emphasis on the forward divergence in the objective of (PI) [... and] (temporarily) break the symmetry of the Jeffreys divergence [...]. Dynamically changing the focus between forward and reverse KL divergences steers training between mode-covering and mode-seeking behaviors”* (line 181-187).
>
>     We believe (together with reviewers j8Dk and rxKB) that the manuscript already motivates the method sufficiently clearly, but we welcome any specific suggestion the reviewer might have to avoid misunderstandings.
>
>
> 2.  **Validity of KL divergence**: While we agree that all divergences are “valid,” note that we compute divergence with respect to the empirical distribution $\hat{\pi} = \frac{1}{N} \sum_{n} \delta(x-x_n)$ (where $\delta$ is the Dirac delta) and not the true underlying $\pi$, which we assume is inaccessible. Perfectly fitting the empirical distribution is useless when trying to approximate $\pi$. What is more, all models, regardless of how flexible, have limited expressivity. The minimum of any divergence will therefore not be zero, leading to different compromises on how we fit the underlying distribution. As explained above, our method adaptively adjusts the different trade-offs arising from the use of the forward and reverse KL.
>
>     Naturally, as we point out in the conclusion, our method can be used with “different models and divergences” (lines 481-482), including $f$-divergences such as those from the work linked by the reviewer. We could not locate a published version of the preprint, but if they provide a citation we can also include it as an example.
>
>
>
>
>
> 3.  **Experimental results**: The reviewer has a point that our image  generation experiments cannot compete with state-of-the-art generative models. However, the method proposed in the paper does not tackle only the problem of sampling, but that of learning a distribution. While the extent to which generative models are capable of doing that is unclear, our method is directly designed to do that. For that reason, it can be used in applications where other generative models cannot, such as density estimation and classification. What is more, while diffusion or VAE models may be able to generate higher-quality images, they cannot compute the probability associated with a sample (even those generated by the model). In contrast, our method is explicitly designed to do so. For this reason, our experiments compare our method to others with similar capabilities (e.g., NF), in which scenario the advantage of our method is clearer.
>
>     That being said, the reviewer has a point that this method could have issues in very high dimensions, a limitation we note in our conclusion. This is connected mainly to the choice of EBM and NF and not necessarily to the training method we propose. We note in the conclusion that an interesting avenue of future works “is to extend this method to [...] [high dimensional settings] by estimating the probability of generated samples (Ben-Dov et al., 2024) or by using rectangular flows (Caterini et al., 2021)” (lines 478-479). We also suggest that this challenge could be addressed by “using different models and divergences in (P-DYN). Strong candidates are score-based models with the Fisher divergence. There is a form of the Fisher divergence that avoids the score of π by computing the Hessian” (lines 482-484). These suggestions, though promising, are beyond the method proposed in the manuscript and we agree with the reviewer’s assessment that papers should have “a new method with good motivation or [...] improve some current best methods.” We consider this work to fall in the former category.

---

> > ### Author Response · Authors · 2025-11-20
> >
> > 4.  **Additional proxy model**: The use of additional models to improve performance is pervasive in ML, sampling, and generative modeling, as illustrated by GANs (the discriminator), reinforcement learning (critic), and VAEs (encoder and decoder models). In all cases, the additional model needs to be sufficiently powerful.
> >
> >     Note, however, that in contrast to GANs, the proxy model $q_\psi$ is useful not only during training. In fact, it is one of the outputs of the method. For that reason, its performance is reported throughout our experiments (see, e.g., Fig. 3, 4, and 6). In doing so, our method effectively decouples two sampling/generative modeling tasks: by simultaneously training an NF ($p_\theta$) and an EBM ($q_\psi$), our method provides a (potentially) more flexible model that is good for sampling (EBM $q_\psi$) and another model capable of evaluating the probability of those samples (NF $p_\theta$).
> >
> >     We’ve included a remark to this effect in the revision (lines 474-475) to make the advantages of our method clearer.
> >
> >
> >
> > 5. **Inspiration**: We respectfully, but strongly disagree with the reviewer that scientific contributions should be judged by whether they are “inspired.” The ICLR guidelines explicitly mention “motivation” (addressed in point 1 and praised by both reviewers j8Dk and rxKB) and not “inspiration.”

---

### Official Review · Reviewer_rxKB · 2025-10-30

**Soundness:** 3
**Presentation:** 3
**Contribution:** 4
**Rating:** 8
**Confidence:** 4

**Summary:**

The paper proposes a new approach to generative modeling by minimizing the Jeffrey's divergence.  The forward KL $D(\pi|p_\theta)$ is trained in the usual MLE way using samples of $\pi$ while the reverse KL $D(p_\theta|\pi)$ which is not directly accesible is trained by using a surrogate distribution $q_\psi$.   Instead of minimizing $D(p_\theta|\pi)$ directly one minimizes instead $D(p_theta|q_\psi)$ (which is then computed "explictly" or via MC) while keeping $D(\pi|q_\psi)$ small.

The main idea is to avoid the adversarial (and sometime brittle) approach to GANs which is use the dual formulation of the KL divergences and neural newtowkr architectures. In the current approach the family $p_\theta$ and $q_\psi$ are parametrized directly a combination of neural flows and energy models to ensure expressivity.

The resulting objective functional is not convex so it is treated by dual optimization and a control on the difference bewteen solutions of the original and dual functional.

**Strengths:**

1) The paper is very well written, the main ideas and concepts are presented with clarity and in a nuanced manner.

2) The ideas in the paper are novel and original.  As far as the reviewers knows,  this is a completely new approach to generative model and a new way to avoid the adversarial training of GANs.   In some way the introduction of a surrogate  replaces the back-forward training commong in flow models (such as diffusion models or normalizing flows)  by the introduction of the surrogate. The combined use of neural flows and energy model is also interesting.

3) The experiments are overall sufficient to demonstrate the effectiveness of the training.

4) The reviewer appreciate the thoughtful discussion  about the limit of the methods in high-dimension, and maybe the need for other divergences

**Weaknesses:**

1)  The fact that generative/surrogate  divergence is handled via importance sampling  MC is a little bit worrying, especially if the target has a complex structure with metastable behavior.

2) The dual optimization framework seems super interesting.  The reviewer would have appreciated a bit more background and intuition about why this works and why this apply here.  In particular the assumption about closeness in total variation (which is a very strong norm) is not very likely to be true in practice.

**Questions:**

My two questions would be to adress the two weakness noted above.

1) Can you explain where your method starts to fail?

2) How can one understand the theory behind the optimization problem better?  It seems from the experiments that that the gap between primal and dual solutions is very small.

---

> ### Author Response · Authors · 2025-11-20
>
> We thank the reviewer for fully engaging with our paper and appreciate that they find our ideas clear and original.
>
> 1.  **Importance sampling for the proxy**: We share the reviewer’s concerns and find that this is one of the main avenues for future work. As we state in our conclusion (lines 479--481): *“while estimating the partition $\zeta_\psi$ using samples from $p_θ$ is relatively efficient (Appendix E), high dimensions may require streamlining this process by adaptively changing the number of samples and the frequency of estimation.”* This could potentially be an issue for challenging distributions, such as the metastable ones mentioned by the reviewer.
>
>     Yet, notice that we do not use importance sampling from $\pi$, but from $p_\theta$. While $p_\theta$ could model a complex distribution, it will remain most of the time benign for computational reasons (e.g., a GMM or NF). Moreover, the divergence between $p_\theta$ and $q_\psi$ is explicitly minimized by the optimization problem, as it is our surrogate for the reverse KL. This has the side-effect of improving the performance of importance sampling techniques and keeping our method computationally efficient and statistically stable (Appendix E). Still, not using enough samples can have negative effects, as illustrated in Figure 8b (Appendix E).
>
>
>
> 2.  **Failure cases**: In view of the discussion above, the main limitation in implementing this method remains dimensionality rather than the underlying distributions (as illustrated in Section 5.3). In high dimensions, computing the partition function with importance sampling requires, at least initially, a large amount of samples. Moreover, the high dimension can lead to big differences in probability values between both models, which leads to exploding exponents. We suggest ways to mitigate these problems in Appendix B.8, but it remains the main avenue for future work (see Conclusion). Another potential failure mode is the vanishing partition function $\zeta_\psi$. This is particularly an issue when using EBMs as proxy models. Due to their unnormalized nature, i.e., since $q_\psi = \exp(-g_\psi(x))/\zeta_\psi$, maximizing their value over the data samples can be achieved by shrinking their partition function $\zeta_\psi$. As the partition function gets closer to 0, the method becomes increasingly numerically unstable. We illustrate this point in Figure 2b which illustrates how the partition function can wildly change during optimization. We address this failure mode by constraining the values of $\zeta_\psi$ (lines 309-310) and carefully setting the learning rates of the corresponding dual variables. As Figure 2b shows, “this constraint effectively controls the value of $\zeta_\psi$” (line 346). These stability issues could also be mitigated by using, e.g., regularization, projection methods, or improved EBM architectures. We included a remark to this effect in the revision section 5.1 (lines 347-348).
>
>
> 3.  **Duality theory**: Duality is indeed a core component of our method: solving the constrained problem (P-DYN) would otherwise be quite difficult as its non-convexity precludes most approaches based on, e.g., projections or barriers. The main idea behind duality in optimization is that we can associate an unconstrained problem D-DYN (the dual) to any constrained problem such as P-DYN (the primal). Since the dual is a relaxation of the primal (hard constraints become violation costs), it is not necessarily a useful problem. But note that, in contrast to the penalty problem in P-W (where weights $w$ are fixed), the “weights” $\lambda$ of D-DYN are optimization variables (known as dual variables). This turns out to be an important distinction since it allows us to give conditions under which P-DYN and D-DYN are, to a large extent, equivalent. This phenomenon is called strong duality. For convex optimization problems, strong duality holds under very mild conditions. What Theorem 1 states is that it (approximately) holds for P-DYN and D-DYN despite their non-convexity. We can therefore use a gradient descent-ascent method (Algorithm 1) to solve P-DYN and adapt its parameters $\epsilon$ automatically.
>
> 	While this result is fundamental to our method, duality theory is beyond the scope of this paper. For that reason, we rely more heavily on recent results from the literature, such as (Chamon, 2023). We refer the reviewer to that source and references therein for more details.

---

### Official Review · Reviewer_j8Dk · 2025-11-01

**Soundness:** 3
**Presentation:** 3
**Contribution:** 3
**Rating:** 6
**Confidence:** 1

**Summary:**

The paper proposes a collaborative alternative to adversarial training for minimizing the Jeffreys divergence. The key idea is to introduce a proxy model \(q_ψ\) that both fits the data and serves to approximate the reverse KL term \(DKL(p_θ || π)\) via \(DKL(p_θ || q_ψ)\).

**Strengths:**

This study proposes an adversarial training method for minimizing the Jeffreys divergence. The method is well motivated and theoretically supported. I find no critical flaws in the derivation. However, I am not very familiar with energy-based models and could not fully appreciate the significance of the contributions.

**Weaknesses:**

See above.

**Questions:**

- As I understand it, the KL divergence is preferred because it is connected to maximum likelihood estimation, which yields asymptotically efficient estimators. What is the statistical advantage of symmetrizing the KL divergence?

---

> ### Author Response · Authors · 2025-11-20
>
> We thank the reviewer for their time and we are glad they find the method well-motivated.
>
> 1.  **Concrete disadvantages of asymmetric KL**:We expand the theoretical discussion in our next answer. Here, we showcase a practical case to illustrate what happens when minimizing the forward or reverse KL. Consider fitting one Gaussian ($p_\theta$, where $\theta = (\mu,\sigma^2)$ collects the mean $\mu$ and variance $\sigma^2$ of the model) to a distribution $\pi$ composed of a mixture of two Gaussian $N_1$ with mean 0 and unit variance and $N_2$ with mean $m$ and unit variance, namely, $\pi=0.5 N_1 + 0.5 N_2$. As soon as these modes are sufficiently apart (e.g., $m = 6$), the behavior of the forward and reverse KL becomes different.
>     Indeed, from the definition of the forward KL $D_{KL}( \pi ∥p_{\theta})$ in (PI), it is clear that it can only be finite if $\pi > 0 \Rightarrow p_\theta > 0$. In other words, it enforces that the whole support of $\pi$ must be covered by $p_\theta$. This leads to solutions that we referred to as “mode-covering” (in this case, $\theta = (\mu,\sigma^2) = (3.5,16)$). On the other hand, for the reverse KL $D_{KL}(p_{\theta} ∥ \pi)$ to be finite, the opposite must holds, i.e., $\pi = 0 \Rightarrow p_\theta = 0$. Hence, it enforces that only the support is covered, but not necessarily all of it. This leads to what is often referred to as “mode-seeking” behavior (in this case, $\theta = (\mu,\sigma^2) = (0,1.3)$). It is worth noting that if both distributions are sufficiently close together, both KL lead to similar, though not identical, results (e.g., for $m = 5$, minimizing the forward KL yields $\theta = (\mu,\sigma^2) = (2.3, 7.1)$, whereas the reverse KL yields $\theta = (\mu,\sigma^2) = (2.7, 5.9)$).  This is also nicely visualized in https://www.tuananhle.co.uk/notes/reverse-forward-kl.html .
>
>
>
> 2. **Statistical advantage of symmetrizing the KL divergence**: As we explain in the paper (lines 131-133), *“the KL divergence is not symmetric, and minimizing the forward KL divergence $D_{KL}(\pi \Vert p_\theta)$, as in (P-KL), leads to solutions that cover the support of $\pi$ at the cost of lower accuracy over sharp modes (Murphy, 2012).”* Indeed, since the entropy of $\pi$ is constant with respect to $\theta$, we see from (P-KL) that it can be optimized as $D(\pi \Vert p_{\theta}) = -E_{\pi} [ \log p_{\theta} (x) ]$. Hence, minimizing the forward KL is akin to performing maximum likelihood estimation and $E_{\pi} [ \log p_{\theta} (x) ]$ is largest when $\pi > 0 \Rightarrow p_\theta > 0$, leading to this support-covering behavior.
> 	Continuing from the paper, “the reverse KL divergence $D_{KL} ( p_{\theta} \Vert \pi )$ promotes a mode-seeking behavior that may ignore regions of $\pi$ with significant mass.” The reason being that, once again from (P-KL), we now have $D_{KL}(p_{\theta} \Vert \pi) = -H(p_{\theta})-E_{p_{\theta}}[\log \pi (x)]$. Hence, $E_{p_{\theta}}[\log \pi]$ is maximized by focusing on a mode of $\pi$ (since $\pi$ does not contribute wherever $p_\theta = 0$). As the manuscript goes on to state, “one way to overcome this asymmetry is combining forward and reverse divergences [...]”.
> 	The advantage of a symmetric divergence becomes apparent when the amount of data is finite and small, which exacerbates the discrepancies described above. Indeed, attempting to cover the support of an empirical distribution may lead to noisy samples getting higher probability than they actually have. On the other hand, mode-covering behavior will tend to assign no probability to regions where there are no samples, even though they may be part of the support of the true underlying distribution. Combined, however, these behaviors allow better coverage of the support of $\pi$, while still taking into account the importance of fitting its modes.
>
>
> 3. **KL inaccuracy and Jeffreys technical difficulty**: The reviewer has a point that the forward KL is computationally easier to minimize than the Jeffreys divergence (more specifically, than the reverse KL). However, we hope that our responses to their points 1 and 2 have shown that the former, while more convenient, may lead to poor fits to the underlying distribution $\pi$. To further drive this point, consider Figure 3a, where we see that the NLL of the NF (trained using only forward KL) diverges. This is the reason behind the many attempts to develop methods that minimize symmetric divergences, such as GANs. This paper uses a different approach that seeks to be more accurate than minimizing only the forward KL (as in Figure 3a) and more stable than the adversarial approach of GANs (Figure 3b). Indeed, there is a natural trade-off between accuracy and computational difficulty and each use case may require a different balance.

---

### Meta-Review · Area_Chair_Mye2 · 2026-01-13

**Summary:**

The paper proposes a method to minimise the Jeffreys divergence, a symmetrised KL divergence, using a proxy model to approximate the reverse KL. Reviewers j8Dk and rxKB found the paper well written, novel and with sufficient experimentation to demonstrate effectiveness. Reviewer rxKB raised some concerns about the need for importance sampling for the proxy. Reviewers sXKM and Nftq raised significant concerns about the motivation, the experimental results, the theoretical justification, and scalability to higher dimensions.

**Reviewer Concerns:**

The rebuttal responds to concerns about the motivation for symmetrising the KL, the limitations of importance sampling, and failure modes. However, the concerns about evidence for high-dimensional behaviour and experimental results remain.

**Reviewer Scores:**

Reviewers j8Dk and rxKB would likely maintain their scores, as they had only minor concerns and their initial positions were already positive. Reviewers sXKM and Nftq would likely also maintain their initial scores, as although the rebuttal responds in detail to the motivation and theoretical justification, it does not alter the reviewers' core objection about weak results and potential issues with scaling to higher dimensions. Given the initial scores, it is unlikely that the overall evaluation would shift sufficiently toward acceptance, even if some reviewers raised their scores. Therefore, I recommend rejecting the paper. However, I highly recommend that the authors submit a revised version of the manuscript to the next ML conference.

---

### Decision · Program_Chairs · 2026-01-26

Reject